# Identification of novel HPFH-like mutations by CRISPR base editing that elevate the expression of fetal hemoglobin

Nithin Sam Ravi[1,2], Beeke Wienert[3,4,5], Stacia K Wyman[3], Henry William Bell[5], Anila George[1,2], Gokulnath Mahalingam[1], Jonathan T Vu[3], Kirti Prasad[1,6], Bhanu Prasad Bandlamudi[1], Nivedhitha Devaraju[1,6], Vignesh Rajendiran[1,2], Nazar Syedbasha[1], Aswin Anand Pai[2,7], Yukio Nakamura[8], Ryo Kurita[9], Muthuraman Narayanasamy[1,10], Poonkuzhali Balasubramanian[2,7], Saravanabhavan Thangavel[1], Srujan Marepally[1], Shaji R Velayudhan[1,2,7], Alok Srivastava[1,2,7], Mark A DeWitt[3,11], Merlin Crossley[5], Jacob E Corn[3,12], Kumarasamypet M Mohankumar[1,2]*

[1]Centre for Stem Cell Research (a Unit of inStem, Bengaluru), Christian Medical College Campus, Vellore, India; [2]Sree Chitra Tirunal Institute for Medical Sciences and Technology, Thiruvananthapuram, India; [3]Innovative Genomics Institute, University of California, Berkeley, Berkeley, United States; [4]Institute of Data Science and Biotechnology, Gladstone Institutes, San Francisco, United States; [5]School of Biotechnology and Biomolecular Sciences, University of New South Wales, Sydney, Australia; [6]Manipal Academy of Higher Education, Karnataka, India; [7]Department of Haematology, Christian Medical College & Hospital, Vellore, India; [8]Cell Engineering Division, RIKEN BioResource Center, Ibaraki, Japan; [9]Research and Development Department, Central Blood Institute Blood Service Headquarters, Japanese Red Cross Society, Japan, Tokyo, Japan; [10]Department of Biochemistry, Christian Medical College, Vellore, India; [11]Department of Microbiology, Immunology and Molecular Genetics, University of California, Los Angeles, Los Angeles, United States; [12]Institute of Molecular Health Sciences, Department of Biology, Zurich, Switzerland

*For correspondence: mohankumarkm@cmcvellore. ac.in

Competing interest: The authors declare that no competing interests exist.

**Abstract** Naturally occurring point mutations in the *HBG* promoter switch hemoglobin synthesis from defective adult beta-globin to fetal gamma-globin in sickle cell patients with hereditary persistence of fetal hemoglobin (HPFH) and ameliorate the clinical severity. Inspired by this natural phenomenon, we tiled the highly homologous *HBG* proximal promoters using adenine and cytosine base editors that avoid the generation of large deletions and identified novel regulatory regions including a cluster at the –123 region. Base editing at –123 and –124 bp of *HBG* promoter induced fetal hemoglobin (HbF) to a higher level than disruption of well-known BCL11A binding site in erythroblasts derived from human CD34+ hematopoietic stem and progenitor cells (HSPC). We further demonstrated in vitro that the introduction of –123T > C and –124T > C HPFH-like mutations drives gamma-globin expression by creating a de novo binding site for KLF1. Overall, our findings shed light on so far unknown regulatory elements within the *HBG* promoter and identified additional targets for therapeutic upregulation of fetal hemoglobin.

## Editor's evaluation

This paper describes the innovative use of base editing to mutagenize an enhancer region in the iconic globin locus, demonstrating a new method while also finding a potential novel locus for downstream therapeutic approaches.

## Introduction

Fetal hemoglobin (HbF) is a tetramer consisting of two alpha-globin chains and two gamma-globin chains, which are highly expressed during the fetal stage of human life. The expression of HbF is silenced progressively after birth until it constitutes only about 1% of total hemoglobin (*Bauer and Orkin, 2012*). Naturally occurring mutations in the regulatory regions of the gamma-globin (*HBG*) genes have been shown to reactivate expression and increase HbF levels during adult life (*Jacob and Raper, 1958*). This inherited genetic condition is benign and is known as hereditary persistence of fetal hemoglobin (HPFH). Individuals, who inherit HPFH alongside other genetic disorders affecting the adult beta-globin gene, such as sickle cell disease or beta-thalassemia, were shown to have fewer, if any, symptoms (*Jacob and Raper, 1958*; *Thein, 2018*). Hence, high levels of HbF expression have been shown to be beneficial for improving the clinical outcomes of patients with sickle cell anemia and beta-thalassemia.

Genome editing approaches have largely focused on the beneficial effects of HPFH mutations to increase HbF levels in sickle cell disease (*Traxler et al., 2016*). These mutations either create de novo binding sites for erythroid activators or disrupt the binding sites of repressors, thereby increasing the expression of HbF. For example, the −175T > C, −198T > C and −113A > G HPFH point mutations create de novo binding sites for the erythroid master regulators TAL1, KLF1, and GATA1, respectively (*Fischer and Nowock, 1990*; *Martyn et al., 2019*; *Stoming et al., 1989*; *Wienert et al., 2017*; *Wienert et al., 2015*). Similarly, the introduction of HPFH-associated mutations around −115 bp from the transcription start site (TSS) of *HBG* (−114C > A, −117G > A, and a 13 bp deletion [Δ13 bp]), and around −200 bp from the TSS (−195C > G, −196C > T, −197C > T, −201C > T and −202C > T/G), were shown to disrupt the binding sites of the two major fetal globin repressors, BCL11A and ZBTB7A/LRF, respectively (*Martyn et al., 2018*). However, the roles and locations of other regulatory elements in the *HBG* promoter that are involved in activation or de-repression are less well understood. Thus, tiling the *HBG* promoter using base editors could unravel molecular mechanisms of human hemoglobin switching and reveal additional point mutations that could be useful for therapeutic gamma-globin upregulation.

Targeted introduction of HPFH mutations into the *HBG* promoter by nuclease-mediated homology-directed repair is relatively inefficient and can result in high rates of random insertions and deletions (indels) through non-homologous end-joining DNA repair pathways (*Cavazzana et al., 2017*). In addition, due to the high homology between the duplicated *HBG1* and *HBG2* genes, simultaneous editing of both *HBG* promoters by programmable nucleases that cause double-stranded breaks (DSBs) sometimes results in ~4.9 kb deletion comprising the *HBG* intergenic region with uncertain consequences (*Li et al., 2018*; *Métais et al., 2019*; *Wienert et al., 2017*).

To overcome these limitations, we implemented a strategy to screen and identify potential regulatory mutations within the proximal promoters of the two human fetal globin genes *HBG1* and *HBG2*. We employed CRISPR base editing to introduce an array of point mutations into the *HBG* promoters and then screen for those mutations that induce HbF to therapeutic levels, without the confounding effects of creating DSBs. Similar to previous findings, we observed that base editing using adenine and cytosine base editors (ABEs and CBEs, respectively) is highly efficient in creating point mutations without inducing high levels of indels (*Gaudelli et al., 2017*; *Komor et al., 2016*). We identified several novel point mutations that are associated with a significant increase in gamma-globin expression and could be of therapeutic interest. Our results demonstrated that base editors are a powerful tool for mapping the so far unknown regulatory elements within the *HBG* promoters and provide a proof-of-concept approach for the treatment of beta-hemoglobinopathies.

## Results

Previous studies have shown that the highly homologous *HBG1* and *HBG2* proximal promoters play a crucial role in the gamma-globin expression. Several non-deletional forms of HPFH-associated point

mutations in the promoter region of *HBG1* and *HBG2* have been associated with increased expression of gamma-globin (*Wienert et al., 2015*). To identify novel regulatory elements in the human *HBG* promoters that influence gamma-globin expression, we performed a base editing screen to introduce point mutations in all compatible locations within 320 bp upstream of the TSS of the *HBG* genes. In brief, we created stable HUDEP-2 cells (*Kurita et al., 2013*) (an immortalized human erythroid progenitor cell line) expressing base editors (ABE or CBE), and then screened guide RNAs (gRNAs) targeting the proximal promoter region of *HBG1* and *HBG2* for their ability to upregulate fetal globin expression. The top gRNAs were validated for editing efficiency and HbF levels. Moreover, the plausible mechanism of novel gRNAs identified from the study on HbF elevation was further characterized by electrophoretic mobility shift assay (EMSA) and Chromatin immunoprecipitation quantitative PCR (ChIP-qPCR). Finally, the potential therapeutic induction of HbF levels for the identified novel gRNAs were validated in erythroid cells derived from healthy donor CD34+ HSPCs.

## Base editors as a preferred genome editing tool for targeting the highly homologous *HBG* promoter region

First, we generated stable HUDEP-2 cell lines that express different gene editors, ABE, CBE, or Cas9, respectively. HUDEP-2 cells were transduced with ABE7.10 RA, BE3RA-FNLS, or Cas9 lentiviral constructs (hereafter named HUDEP-2-ABE, CBE, or Cas9). The vector copy number (VCN) of HUDEP-2 cells transduced with ABE, CBE, or Cas9 lentiviral constructs ranged from 0.25 to 0.85 by real-time PCR (*Figure 1—figure supplement 1a*). Previously defined sgRNAs targeting the BCL11A binding motif (*Traxler et al., 2016*) in the *HBG1* and *HBG2* promoters with a suitable editing window for ABE, CBE, and Cas9 were transduced with the VCN of 0.6–1.2 (*Figure 1—figure supplement 1a*). The editing efficiency was 88% for Cas9, 10–51% for ABE, and 59–73% for CBE, with the transduction efficiency (as measured by GFP expression) greater than 98% (*Figure 1a*). After differentiating the cells into erythroid progenitors, the percentage of HbF positive cells was higher in case of ABE and CBE than Cas9 (*Figure 1b*). While gene editing was very high with Cas9, we did not observe a corresponding increase in the HbF positive cells which might be due to a previously described 4.9 kb deletion comprising the *HBG2* gene and *HBG1-HBG2* intergenic region. It has been previously reported that introducing DSBs in highly homologous *HBG* promoters with Cas9 nucleases may generate this deletion (*Li et al., 2018*). Therefore, we determined the frequency of this deletions by qRT-PCR in all the edited samples. As expected, the 4.9 kb deletion was observed at a high frequency (76%) in Cas9 edited cells. We also noted some deletions in base edited samples but significantly fewer than with Cas9 (*Figure 1c*). Consistent with the frequency of the 4.9 kb deletion, the globin chain analysis by RP-HPLC showed lower levels of G gamma but not A gamma chain only in Cas9 edited cells in comparison to ABE and CBE edited cells after normalizing to the control (*Figure 1d*). These results suggest that ABE and CBE are highly efficient in editing the highly homologous regions like gamma-globin promoter without causing a large deletion between the two *HBG* genes.

## Screening of *HBG* proximal promoter with base editors identifies novel HPFH like mutations

To identify the potential regulatory regions involved in gamma-globin expression, we therefore selected ABE and CBE for tiling the *HBG* promoter. Transcriptomic analysis of the stable cell lines expressing ABE and CBE showed a significant correlation with the wild type HUDEP-2 cells, confirming that the gene expression profiles are not altered (*Figure 1—figure supplement 1b*). As the base editors and gRNAs are constitutively expressed, we determined the editing frequency of ABE and CBE stables with gRNA-2 for its effect on HbF elevation at different time points during expansion and differentiation. The editing efficiency and HbF levels in both ABE and CBE increases over time with no discernable effect on erythroid differentiation (*Figure 1—figure supplement 1c-h*). We then generated ABE and CBE gRNAs in all compatible locations up to 320 bp upstream of the TSS of the *HBG1* and *HBG2* promoters. Guide RNAs were designed with a suitable base editing window (target nucleotide in positions 3–9 from NGG PAM distal end) for ABE and CBE (*Figure 2a*). Among the 41 gRNAs designed, 36 gRNAs had a base editing window for ABE, and 32 gRNAs had a base editing window for CBE (*Supplementary file 1*). An overview of the methodology used in this study is illustrated in *Figure 2b*: All the gRNAs were cloned in a lentiviral vector with a GFP reporter. Lentivirus was produced for each gRNA; HUDEP-2-ABE and -CBE cells were then transduced in an arrayed format

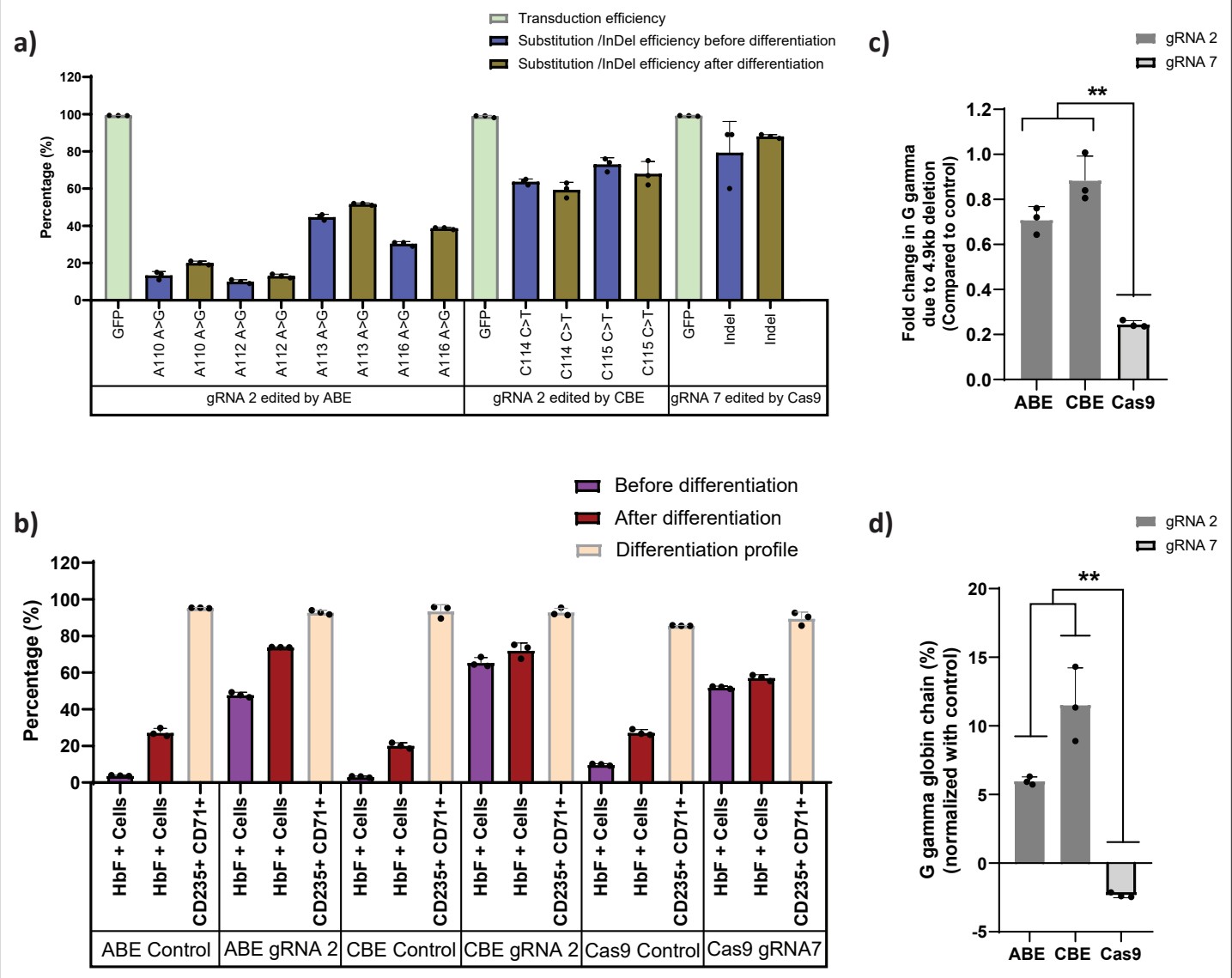

**Figure 1.** Base editors are preferred tool over Cas9 for editing the highly homologous *HBG1* and *HBG2* promoter. Highly homologous *HBG* promoter was edited by adenine base editor (ABE), cytosine base editor (CBE), and Cas9 with suitable guide RNAs (gRNAs) that target the well-known BCL11A binding site (–115 transcription start site [TSS]). (**a**) Transduction efficiency of gRNA-2 (for ABE and CBE) or gRNA-7 (for Cas9), percentage of individual base conversion for ABE and CBE (with gRNA-2) and insertions and deletions (indels) for Cas9 (with gRNA-7) before and after erythroid differentiation are represented. The transduction efficiency was analyzed by FACS, the individual base substitution and indel percentage were analyzed by EditR and ICE software respectively after sanger sequencing. (**b**) Flow cytometry analysis of fetal hemoglobin (HbF) and erythroid maturation markers (CD235a and CD71) expression in edited HUDEP-2 cells. The percentage of HbF-expressing cells were analyzed before and after differentiation into erythroblasts. (**c**) Analysis of *HBG2* deletion (due to 4.9 kb deletion) by qRT-PCR in the base edited and Cas9 edited HUDEP-2 cells. (**d**) Expression of G gamma-globin chain in ABE, CBE, and Cas9 edited HUDEP-2 cells, measured by RP-HPLC after differentiation into erythroblasts. The data were normalized with respective controls. Data are expressed as mean ± SEM from three biological replicates, asterisks indicate levels of statistical significance (**p < 0.01).

The online version of this article includes the following figure supplement(s) for figure 1:

**Figure supplement 1.** Characterization of the HUDEP-2 cells expressing adenine (ABE) and cytosine base editor (CBE).

with equal transduction efficiency (~1 VCN/cell). The mean transduction efficiency for all these gRNAs in both ABE and CBE samples were around 97%. The gRNA transduced cells were then expanded for 8 days, successful base editing was then confirmed by NGS and Sanger sequencing.

First, we determined the overall efficiency of ABE-induced A-to-G conversions and CBE induced C-to-T conversions at different target sites of *HBG* promoter by NGS. The gRNAs associated with lower base editing efficiency (<10%) were excluded from further analysis as they do not provide

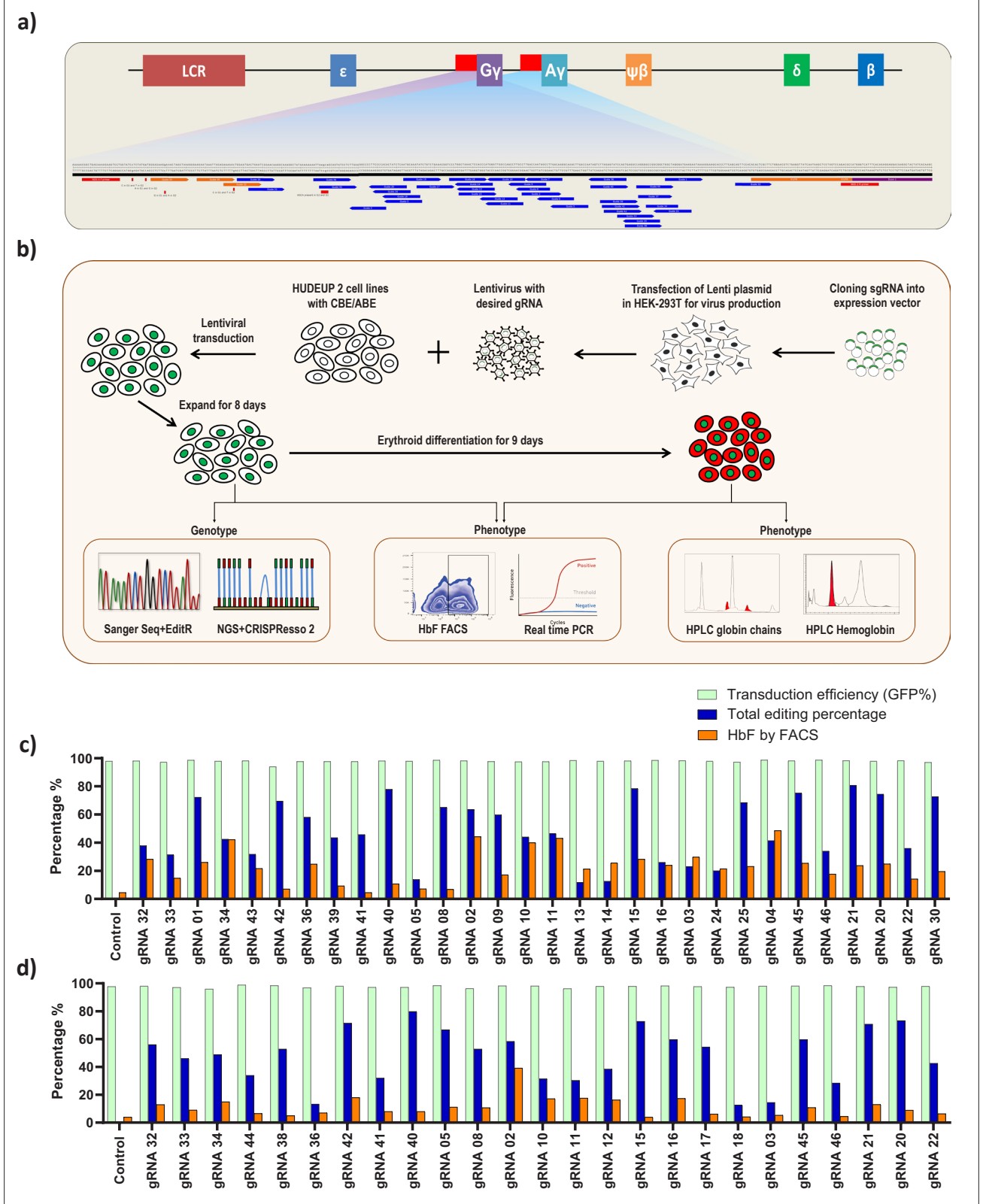

**Figure 2.** Screening of *HBG* promoter using base editors to identify novel point mutations that elevate fetal hemoglobin (HbF) expression. (**a**) Schematic representation of the overall screening approach, adenine base editor (ABE) or cytosine base editor (CBE) expressing HUDEP-2 cells were transduced with guide RNA (gRNAs) that target the proximal promoter of the *HBG* gene. The edited cells were expanded for 8 days. Editing efficiency was evaluated by Sanger sequencing and NGS, while functional analysis was carried out using FACS and qRT-PCR. Top targets from both the ABE

*Figure 2 continued on next page*

*Figure 2 continued*

and CBE screens with the highest induction of HbF were validated and differentiated to erythroid cells. The differentiated cells were further subjected to FACS, qRT-PCR, RP-HPLC, and HPLC analysis to determine the number of HbF positive cells, *HBG* expression, individual gamma-globin chains, and fetal hemoglobin levels, respectively. (**b**) Representation of gRNA targeting *HBG* promoter region in HUDEP-2 cell line, gRNAs targeting –320 bp upstream of transcription start site (TSS) in *HBG* genes (*HBG1* and *HBG2*) promoter regions are represented in the figure. gRNAs common for *HBG1* and *HBG2* promoters are represented in blue, while the gRNAs specific to *HBG1* promoter are represented in orange color, the primers used for deep sequencing are represented as a red bar. Comparison of transduction efficiency, base editing frequency, and HbF expression in HUDEP-2 cells expressing ABE (**c**) and CBE (**d**) transduced with different gRNAs targeting the *HBG* proximal promoter. The base edited cells were sequenced by NGS and analyzed for total editing frequency using CRISPResso-2. The transduction efficiency (GFP+ cells) and HbF positive cells were analyzed by FACS.

The online version of this article includes the following figure supplement(s) for figure 2:

**Figure supplement 1.** Analysis of base substitution efficiency at single base pair resolution in *HBG* promoter by adenine base editor (ABE) and cytosine base editor (CBE) through NGS and Sanger sequencing.

**Figure supplement 2.** The product purity and preferred editing window of adenine (ABE) and cytosine base editors (CBE) at the target site.

**Figure supplement 3.** Base editing of *HBG* promoter to identify nucleotide substitutions that suppress fetal hemoglobin (HbF) expression.

insights on *HBG* regulation (gRNAs -37, -38, -7, -18, -19, -29 in ABE, and gRNAs -1, -35, -37, -6, -7, -13, -19 in CBE) (data not shown). After excluding the low editing gRNAs, the total base editing efficiency (overall conversion achieved by a gRNA) varied from 12% to 81% and 13% to 80% for ABE (n = 30) and CBE (n = 25) respectively (*Figure 2c and d*) as determined by CRISPResso-2 analysis. The individual base conversion frequency (base conversion at single base pair resolution) of ABE (A:T to G:C) and CBE (C:G to T:A) ranged from 0% to 74% and 0% to 61%, respectively (*Figure 2—figure supplement 1a and c*). Sanger sequencing data analyzed by EditR further confirmed the base substitution efficiency at the target loci (*Figure 2—figure supplement 1b and d*). The base substitution efficiency for each gRNAs varied drastically depending on the base editing window for ABE and CBE. The average editing efficiency observed was high (>30%) in the canonical positions for ABE (A5-A7) and CBE (C5-C7), while it varied between 1% to 27% in the non-canonical positions for ABE (A1-A4, A8-A12) and CBE (C1-C4, C8-C16) for the different gRNAs used in this study (*Figure 2—figure supplement 2e-f*).

Subsequently, we explored the product purity and indel percentage for all gRNAs in ABE and CBE, as previous studies have shown that the base editors generate a low frequency of unintended edits at the target sites (*Koblan et al., 2018*). Most of the gRNAs in CBE transduced cells showed unanticipated C- to non-T edits (C-R/G-Y), among which C-to-G conversion was predominant. In the case of ABE, we observed a minimal level of unexpected base conversions (A-Y/T-R) at a few on-target sites, consistent with previous studies (*Gaudelli et al., 2017*; *Komor et al., 2016*, *Figure 2—figure supplement 2a-b*). The indel frequency obtained from deep sequencing data was less than 2% in both ABE and CBE (*Figure 2—figure supplement 2c-d*). Our results suggest that ABE exhibits higher product purity and lower indel frequency than CBE in all cases. In summary, we show that both ABE and CBE can effectively introduce A-to-G and C-to-T nucleotide substitutions respectively in the proximal promoters of *HBG1* and *HBG2*.

To evaluate whether the targeted base substitution at the *HBG* promoter by using ABE or CBE has increased HbF expression, we analyzed the HbF positive cells in ABE and CBE edited cells by flow cytometry after intracellular HbF staining. The percentage of HbF positive cells ranged from 2% to 44% in ABE and 1% to 35% in CBE (*Figure 2c–d*). In our preliminary analysis, among the 30 gRNAs in ABE and 25 gRNAs in CBE that we have screened for editing the *HBG* promoter region, five gRNAs in ABE and one gRNA in CBE showed a greater increase in the number of HbF positive cells (in a range of 40–50%).

We identified several gRNAs in ABE (gRNA -39, -41, -42, -08) and in CBE (gRNA -33, -44, -38, -41, -40, -15, -17, -46, -20, -21) which have higher total editing efficiency (>40%) at the target site but resulted in low HbF level (<10%). We were curious to know whether these gRNAs can affect the binding sites of activators resulting in downregulation of gamma-globin expression. To test this hypothesis, we transduced the selected gRNAs in K562 cell lines stably expressing ABE or CBE, a cell model which has high basal level of HbF expression. The transduction efficiency was more than 98% and achieved a higher individual base editing efficiency for each of the gRNAs in both ABE and CBE (*Figure 2—figure supplement 3a-b*). However, we did not observe any decrease in the number of HbF positive cells (98% of cells were HbF positive) in any of the samples suggesting that the targeted

regions did not have binding sites for essential transcriptional activators (*Figure 2—figure supplement 3c-d*).

Interestingly, some of the top candidates from the screen include target regions that were previously identified as binding sites for BCL11A (gRNA-2), KLF-1 (gRNA-4), and TAL-1 (gRNA-3), but we also identified a few other novel target sites (gRNA-10, gRNA-11, gRNA-15, gRNA-16, gRNA-21, gRNA-32, gRNA-34, gRNA-42). The gRNAs-2, -3, and -4 recreates the well-known naturally occurring HPFH mutations –114C > T, –117G > A, −175T > C and −198T > C (*Liu et al., 2018*; *Martyn et al., 2018*; *Stoming et al., 1989*; *Wienert et al., 2018*; *Wienert et al., 2017*). We compared the percentage of HbF positive cells with the editing efficiency for each of gRNAs at the target region in both ABE and CBE cells (*Figure 2c–d*). The total base editing efficiency was generally higher when compared to the proportion of HbF positive cells except in few cases (gRNA-3, -4, -13, -14, -37, and -38 in ABE edited cells). Together, the candidate gRNAs which upregulated HbF from the primary screening of the *HBG* promoter by ABE and CBE provides targets for further validation.

## Base editing at potential target sites in the *HBG* promoter substantially induces HbF expression

The top eight gRNAs from the ABE screen (gRNAs -2, -3, -4, -10, -11, -15, -32, and -34) and the CBE screen (gRNAs -2, -10, -11, -16, -21, -32, -34, and -42) which resulted in the highest levels of HbF positive cells were further validated. Out of the top eight gRNAs identified from the base editor screen, five gRNAs (gRNA-2, gRNA-10, gRNA-11, gRNA-32, and gRNA-34) were common in both ABE and CBE, indicating that these target regions might play an important role in *HBG* silencing. The edited cells were cultured in erythroid differentiation media after the initial expansion, and a set of functional assays were carried out (*Figure 2b*). Corresponding to the screening results, the total editing efficiency ranged from 24% to 78% and 36% to 85% with mean transduction efficiencies of 96% and 90% for ABE and CBE, respectively (*Figure 3a–b* and *Figure 3—figure supplement 1f*). We observed individual base conversion of A- to-G (ranging from 0% to 65%) or C- to -T (ranging from 1% to 57%) at the respective target regions with less than 2% indel frequency (*Figure 3—figure supplement 1a-b and e*). Further, we also observed the undesired non-C-to-T conversions (i.e., C- to-A or C-to-G) at the on-target site by CBE but not with ABE (*Figure 3—figure supplement 1a-b* and *Figure 3—figure supplement 2a-b*). The distribution of specific nucleotide substitution mediated by ABE or CBE for all the top eight gRNAs are highlighted in *Figure 3—figure supplement 2a-b*, respectively. ABE showed higher base editing efficiencies of the cognate A and Ts (A113 and A116 for gRNA-02, T175 for gRNA-03, T198 for gRNA-04) than the bystander A and Ts (A110 and A112 for gRNA-02, T181 for gRNA-03, T199 for gRNA-04) for the creation of HPFH mutations. In the case of CBE, we also observed the C-to-T base conversion at the nucleotides adjacent to the protospacer sequence as previously observed (*Arbab et al., 2020*; *Webber et al., 2019*). One such example is gRNA-10 and gRNA-11 in CBE; we observed the base conversion outside the protospacer sequence (–117 site) in addition to on-target editing at –122 site within the base editing window (*Figure 3—figure supplement 1b* and *Figure 3—figure supplement 2b*). The base conversion at –117 site disrupts the core binding motif of the major fetal globin repressor – BCL11A (*Martyn et al., 2018*; *Wienert et al., 2018*; *Yang et al., 2019*). We distinguished the editing frequency in *HBG1* and *HBG2* promoters by phasing the edits with single nucleotide variations at positions –271, –307, –317, and –324 which are unique in *HBG1* and *HBG2* promoters, using Bowtie 2 and IGV software (*Robinson, 2012*; *Langmead and Salzberg, 2013*). Our analysis showed that base editing rates were highly similar and there is no variation in base substitution efficiency between the highly homologous *HBG1* and *HBG2* promoters (*Figure 3—figure supplement 1c-d*).

We analyzed the *HBG* expression before and during differentiation by qRT-PCR. We observed a significant increase in the *HBG* mRNA expression for all the top eight gRNAs in ABE edited cells (p < 0.01 - p < 0.0001) (*Figure 3—figure supplement 3a*). In the case of CBE, gRNAs -2, -10, -11, and-42 showed a substantial increase in *HBG* mRNA expression (p < 0.05 - p < 0.0001), while gRNAs -16, -21, -32, and -34 showed a modest level of expression as compared with the control (*Figure 3—figure supplement 3g*) before differentiation. The globin mRNA expression pattern in both ABE (*Figure 3—figure supplement 3b*) and CBE (*Figure 3—figure supplement 3h*) edited cells also followed a similar trend during erythroid differentiation. We also determined the number of HbF positive cells before and after erythroid differentiation using FACS. As expected, the percentage of HbF positive

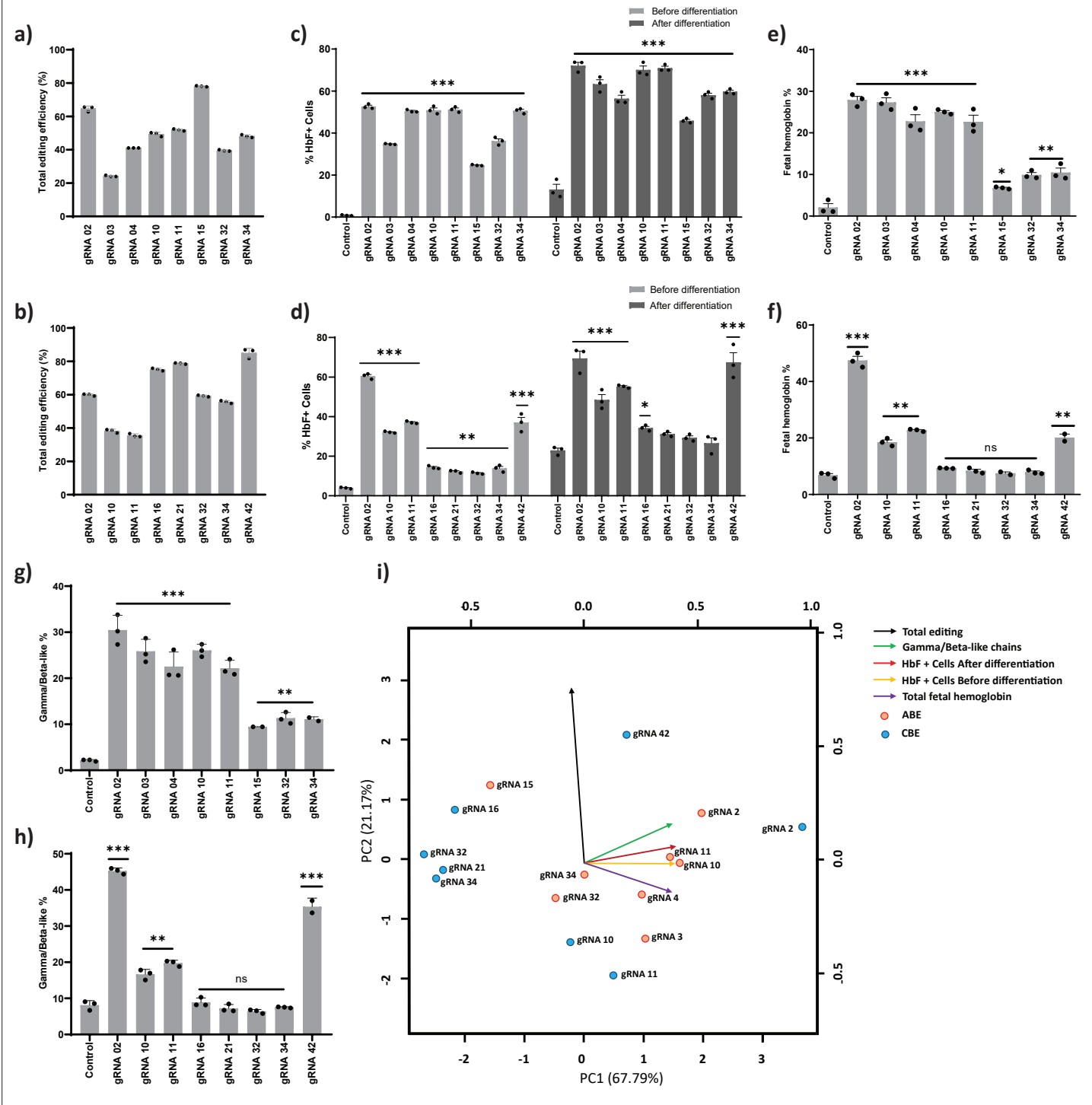

**Figure 3.** Validation of targeted base editing for top eight guide RNAs (gRNAs) from the primary screen of adenine base editor (ABE) and cytosine base editor (CBE) at *HBG* promoters. HUDEP-2 cells expressing ABE or CBE transduced with the top eight gRNAs were analyzed by deep sequencing at the targeted regions in the *HBG* promoter. The total editing efficiencies of ABE (**a**) or CBE (**b**) are represented as the percentage of total sequencing reads with target C:A converted to T:G at specified sites. Evaluation of fetal hemoglobin (HbF) positive cells in HUDEP-2 cells expressing ABE (**c**) or CBE (**d**) transduced with the respective gRNAs, before and after differentiation by flow cytometry; globin chains analysis in ABE (**e**) or CBE (**f**) edited HUDEP-2 cells after erythroid differentiation by RP-HPLC; HbF analysis in ABE (**g**) or CBE (**h**) edited HUDEP-2 cells after erythroid differentiation by HPLC. (**i**) Principal component analysis plot for the correlation between the outcomes of base editing using top eight gRNAs. The relationship between the base edit frequency, HbF+ cells, HbF, and gamma/beta-like chains in ABE or CBE edited HUDEP-2 stable cell for the indicated gRNAs were analyzed.

*Figure 3 continued on next page*

*Figure 3 continued*

The first two principal components are plotted, and the variance accounted for by each principal component is shown. Data are expressed as mean ± SEM from three biological replicates (p > 0.05). Asterisks indicate levels of statistical significance **p < 0.01, ***p < 0.001.

The online version of this article includes the following figure supplement(s) for figure 3:

**Figure supplement 1.** Assessment of base editing efficiency at the highly homologous *HBG1* and *HBG2* promoter.

**Figure supplement 2.** Summary of alleles frequency for top eight guide RNAs (gRNAs) at target site by adenine base editor (ABE) and cytosine base editor (CBE).

**Figure supplement 3.** Adenine and cytosine base editing of *HBG* promoter on globin chain mRNA and protein expression.

**Figure supplement 4.** Evaluation of the low efficiency guide RNAs (gRNAs) that induced fetal hemoglobin (HbF) with hyperactive variant adenine base editor (ABE)8e .

cells in differentiated erythroid cells was slightly higher than that of the undifferentiated edited cells (*Figure 3c–d*). Further, we determined the effect of base editing on erythroid differentiation using flow cytometry analysis with CD235a and CD71 markers. The shift in expression of CD71 positive cells alone to CD71/CD235a double positive cells reflects the erythroid differentiation pattern of HUDEP-2 cells (*Kurita et al., 2013*). The percentage of double positive cells was 83–90% for CBE and above 95% for ABE edited cells compared to the control, which was 77% and 97%, respectively, suggesting that the differentiation ability of the edited cells was not affected (*Figure 3—figure supplement 3d* and *Figure 3—figure supplement 3j*).

Furthermore, the level of globin chains was analyzed by using reverse-phase HPLC in differentiated erythroid cells from both ABE (*Figure 3—figure supplement 3c*) and CBE edited samples (*Figure 3—figure supplement 3i*). We observed a significant induction of gamma-globin chain expression, which represented 10% to30% of total beta-like globin content in all the ABE edited samples (gRNAs -2, -3, -4, -10, -11, -15, -32, and -34) (*Figure 3g*). In the case of CBE, the gamma-globin chain levels were around 6% to45% of total beta-like globin content, among which gRNAs -2, -42, -10, and -11 showed significant elevation when compared to the control (*Figure 3h*). In both ABE and CBE edited cells, the increase in gamma-globin chains was consistently associated with a reciprocal reduction in beta-globin chains thereby maintaining the alpha to beta-like globin chain ratio (*Figure 3—figure supplement 3c* and *Figure 3—figure supplement 3i*). Even though the *HBG1* and *HBG2* promoters were base edited with equal efficiency, *HBG1* showed moderately higher expression levels compared to *HBG2* in most of the ABE and CBE edited cells. The decrease in *HBG2* expression in these samples might be due to biased HbF regulation or the 4.9 kb deletion that deletes the *HBG2* gene.

To find whether decrease in *HBG2* expression is due to the 4.9 kb deletion, gRNAs which showed significant reduction in *HBG2* over *HBG1* expression in ABE (gRNA-4, -10, -11, -15, -32, -34) and CBE (gRNA-2, -10, -11, -32, -34, -42) edited cells were further investigated by qRT-PCR. Interestingly, the frequency of 4.9 kb deletion in ABE ranged from 2% to 32% while in CBE it ranged from 0% to 12% (*Figure 3—figure supplement 3e* and *Figure 3—figure supplement 3k*). To determine the correlation between the reduction in *HBG2* chain expression and the frequency of large deletion, Pearson correlation analysis was performed in the above-mentioned gRNAs in ABE and CBE edited cells. We observed a high correlation (*r* = 0.71) in the case of ABE, whereas much lower correlation was observed with CBE (*r* = 0.26) (*Figure 3—figure supplement 3f* and *Figure 3—figure supplement 3i*). These data suggest that the reduction in G gamma chain expression is due to higher frequency of deletions in the ABE edited samples, while in the case of CBE, the decrease in the G gamma chain expression is independent of larger deletions and might be due to the biased expression of gamma-globin.

We observed a substantial difference in deletion rates across gRNAs as well as between base editors (*Figure 3—figure supplement 3e* and *Figure 3—figure supplement 3k*). The difference in the DNA sequence composition which often affects the editing efficiency might also be responsible for varied deletion observed across the gRNAs targeting the *HBG* promoter. On the other hand, processivity of the editors could account for the difference in deletion observed with the same gRNA while editing with CBE and ABE. As the base editors cannot dock on an already edited strand, CBE with a higher rate of editing (~50% editing on day 1) is prevented from interacting again with the DNA, thus reducing the chances of deletion compared to ABE which takes longer to achieve similar editing (~50% editing on day 8) (*Figure 1—figure supplement 1b and c*). This observation is further supported by the minimal deletion seen in samples edited with ABE8e which has a higher processivity

(~90% editing within 24 hr) compared to both CBE and ABE 7.10 (*Figure 3—figure supplement 4c*, *Richter et al., 2020*).

Next, we analyzed the level of hemoglobin tetramers in the differentiated cells by HPLC to determine whether increase in *HBG* chain expression resulted in functional HbF production. We observed a significant induction of HbF in all the top eight target gRNA transduced cells in ABE (gRNAs- 2, -3, -4, -10, -11, -15, -32, and -34), whereas only gRNA-2, -10, -11, and -42 expressed higher levels of HbF in CBE edited cells (*Figure 3e–f*). Consistent with globin chain analysis, we observed that the increase in HbF variant is associated with compensatory downregulation of adult hemoglobin levels in the edited cells. The relationship between the editing efficiency and HbF expression was analyzed for all the top-scoring gRNAs in ABE and CBE edited cells (*Figure 3i*). Among the validated top eight gRNAs in ABE and CBE, gRNA-2, -10, -11 with ABE and gRNA-2 with CBE resulted in a high target editing efficiency with a corresponding increase in HbF expression. In case of gRNA-42 with CBE, only a modest level of HbF elevation was achieved even with higher editing efficiency. On the other hand, gRNAs -3 and -4 with ABE and gRNAs -10 and -11 with CBE showed higher elevation of HbF levels despite lower base conversion efficiency. The higher number of HbF positive cells with minimal base editing might be due to heterogenous editing at target site per cell since there are two copies each of *HBG1* and *HBG2*. Further, irrespective of editing at the target region, the binding of CRISPR-Cas9 complex through the gRNAs at the *HBG* promoter might disrupt the binding of major transcriptional repressor that are involved in globin expression (*Shariati et al., 2019*).

Among the samples which resulted in higher HbF induction with lower editing efficiency, we validated gRNA-03 and -11 with a hyperactive variant of ABE (ABE8e) to determine whether further elevation in HbF level can be attained by increasing the editing efficiency. The HUDEP-2 cells stably expressing ABE8e were transduced with gRNA-03 and -11, with a VCN of 0.28% for the editor and 0.56% for the gRNA (*Figure 3—figure supplement 4e*). We observed a high percentage of base substitution at the target site (more than 95%) with the corresponding increase in the HbF positive cells and gamma-globin chains in both the gRNAs (*Figure 3—figure supplement 4a-c*). The erythroid differentiation capacity of the edited cells was equivalent to that of control (*Figure 3—figure supplement 4b*). The frequency of larger deletions was also significantly reduced perhaps because of the higher processive rate of ABE8e (*Figure 3—figure supplement 4d*). Thus, the ABE8e variant has improved the base editing efficiency at the target region and provided higher level of HbF induction with a reduced frequency of larger deletion.

Through this screen, we identified multiple novel individual regulatory regions and validated well-known HPFH mutations in the *HBG* proximal promoter that are important for gamma-globin regulation. Interestingly, gRNA-2 (with ABE or CBE) and gRNA-4 (with ABE) disrupt the binding site for the major gamma-globin repressors, BCL11A and LRF/ZBTB7A, and generate the binding motif for the transcriptional activators, GATA1 and KLF1, thus resulting in overall activation of *HBG* expression. Base editing of –114C > T, –115C > T and –116A > G mutations disrupts the binding of BCL11A and the base conversion at –198T > C and –199T > C affect the binding of LRF/ZBTB7A to the *HBG* promoter. Further, the installation of –113A > G and –198T > C mutations by gRNA-2 and gRNA-4, generate a binding site for GATA1 and KLF-1, respectively. Moreover, the base conversion at –175T > C by gRNA-3 (with ABE) creates a TAL1 binding site. The novel gamma-globin regulatory region identified includes the target base substitution mediated by gRNA-10, -11, -15, -21, -32, and -34 with ABE or CBE. With gRNAs -10 and -11, ABE converts nucleotide at –123T > C and –124T > C position, whereas CBE converts nucleotide at –122G > A (on-target editing site) and –117G > A (outside the editing window) positions. Target base substitutions at –123T > C and –124T > C positions result in greater induction of *HBG* expression, equivalent to that of the known HPFH mutations that disrupt the binding of BCL11A. Overall, the potential gRNAs include gRNA-2, 3-, -4, -10, and -11 with ABE and gRNA-2 with CBE that exhibit high induction of HbF expression.

## Base editing of the –123 region of *HBG* promoter in human CD34+ HSPCs

To further determine the therapeutic potential of novel targets identified from this study on induction of gamma-globin expression, we performed base editing of CD34+ HSPCs from healthy donor (*Figure 4a*). Electroporation of the ABE8e mRNA with gRNA targeting the BCL11A binding site (gRNA-2) and –123 novel cluster (gRNA-11) effectively generated highly efficient base editing at the

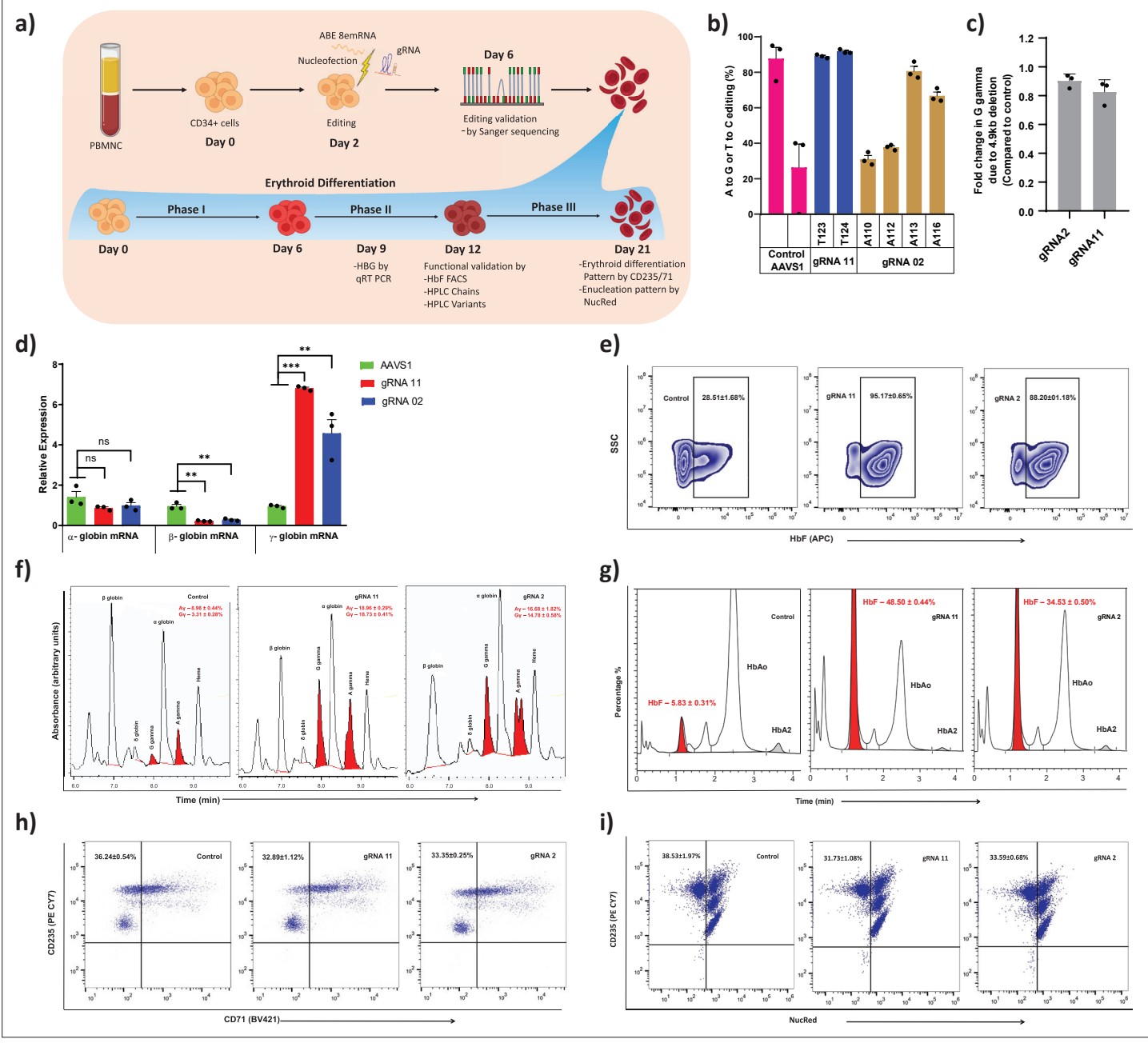

**Figure 4.** Therapeutic induction of fetal hemoglobin (HbF) in erythroblast derived from healthy donor CD34+ hematopoietic stem and progenitor cells (HSPCs) upon base editing of *HBG* promoter. (**a**) Schematic representation of steps involved in based editing of CD34+ HSPCs. Mobilized CD34+ HSPCs from healthy donor were nucleofected using MaxCyte system with adenine base editor (ABE)8e mRNA and respective guide RNAs (gRNAs) on day 2 of expansion. During expansion, CD34+ HSPCs were analyzed at day 6 for the editing efficiency and 4.9 kb deletion. (**b**) Efficiency of individual base conversion at the target sites were measured by EditR after Sanger sequencing. (**c**) Analysis of *HBG2* deletion (due to 4.9 kb large deletion) by qRT-PCR. The based edited CD34+ HSPCs were cultured in a three-phase liquid culture system for erythroid differentiation and enucleation. (**d**) Relative expression of globin transcripts analyzed by qRT-PCR (ΔΔCT) in erythroblasts derived from base edited CD34+ HSPCs on day 9 of differentiation. The functional validation of HbF elevation was analyzed in erythroblasts derived from the indicated samples by FACS, HPLC, and RP-HPLC on day 12 of erythroid differentiation. (**e**) HbF positive cells analyzed by flow cytometry are represented as zebra plots. (**f**) RP-HPLC chromatogram profiles of individual globin chains and (**g**) HPLC chromatogram profile of hemoglobin variants. On the final day of erythroid differentiation, the expression of maturation markers and enucleation fraction were measured by FACS analysis. (**h**) Flow cytometry for the erythroid maturation markers CD235a+ and CD71+. (**i**) Enucleation pattern was determined by flow cytometry analysis for CD235a with NucRed in erythroid cells derived from CD34+ HSPCs. Asterisks indicate levels of statistical significance **$p < 0.01$, ***$p < 0.001$.

target site. The editing efficiency observed at individual base positions were –110 (31%), –112 (37%), –113 (80%), and –116 (66%) with gRNA-2 and –123 (89%), –124 (91%) with gRNA-11 (*Figure 4b*). In case of gRNA-11, the base editing events generated a high proportion of –123 and –124 mutations in combination at the target site. We cultured the base edited CD34+ HSPCs under erythroid differentiation conditions and analyzed HbF expression. The relative levels of *HBG* expression were significantly higher in gRNA-11 (>6-fold) and gRNA-2 (>5-fold) edited samples when compared to control (AAVS1 edited sample) by qRT-PCR (*Figure 4d*). In contrast, a significant downregulation of *HBB* and unchanged levels of *HBA* expression were observed in both the tested targets. Similarly, we observed a substantial increase in HbF protein expression in erythroblast derived from base edited CD34+ HSPCs. Flow cytometry and HPLC variant analysis confirmed the robust increase in the proportion of HbF positive cells and their HbF content compared with control samples for all the tested targets, with the higher effect in gRNA-11 (*Figure 4e and g*). The globin chain analysis showed an increase in expression of *HBG1* and *HBG2* globin chain levels and a reduction of *HBB* globin chain level (*Figure 4f*). Importantly, base editing of the *HBG* proximal promoter with gRNA-2 or -11 did not alter enucleation potential or the expression of erythroid maturation markers CD235a or CD71 (*Figure 4h–i*). Finally, we determined the frequency of the 4.9 kb deletion in CD34+ HSPCs electroporated with ABE8e and gRNA-2 or -11. We observed a very minimal frequency of the 4.9 kb deletion which might be due to higher processivity and transient expression of the base editor mRNA (*Figure 4c*). The present results suggest that the level of HbF induction mediated by the installation of novel –123 cluster HPFH-like mutations (through gRNA-11) is comparable to the naturally occurring –115 cluster HPFH mutations (through gRNA-2) that disrupt the binding site of BCL11A. Together, our data demonstrate that adenine base editing of the *HBG1* and *HBG2* promoters to recreate the novel –123 cluster HPFH-like mutations is a potential approach for the therapeutic induction of fetal globin level and treatment for beta-hemoglobinopathies.

## The –123 T>C and –124 T>C HPFH-like mutations creates a de novo binding site for KLF1

Finally, we investigated the possibility that novel HPFH-like mutations introduced by the base editor might either create or disrupt the binding site for transcriptional regulators. Interestingly, we observed that the base editing at –123T > C and –124T > C sites by ABE with a single gRNA creates the consensus binding site for the master erythroid transcription factor KLF1 (*Figure 5a and b*; *Tallack et al., 2010*). We performed EMSA to verify binding of KLF1 to a probe containing this core element. We observed modest but clear binding of KLF1 to the –123T > C and –124T > C mutated probe in EMSA but not with the wild type probe (*Figure 5—figure supplement 1a-b*) or probes containing either –123T > C or –124T > C mutations alone (*Figure 5c–d*). This confirms that the combination of –123T > C and –124T > C mutation is important for the KLF1 binding to the *HBG* promoter. Next, we performed ChIP experiments to determine whether the KLF-1 directly interacts with –123T > C and –124T > C mutated region of *HBG* promoter in vivo. The KLF1 ChIP was performed in three independent HUDEP-2 clones sorted from wild type cells or the double mutant edited cells, respectively. The ChIP results were normalized to an unrelated positive control, a KLF1 binding site at the SP1 locus. We observed a weak increase in the signal of KLF1 binding to the *HBG* promoters in the cells edited to contain the –123 and –124 mutations, but the effect was modest, and a similarly weak enhancement was also observed at an arbitrary negative control locus, VEGF (*Figure 5—figure supplement 1c*). Thus, as seen in the EMSA, the KLF1 binding is at best weak and may be below the level of detection by ChIP. Future investigations would be required to confirm that KLF1 binding to this site is the main in vivo mechanism of –123T > C and –124T > C HPFH driven upregulation of gamma-globin.

## Off-target and gene expression analysis after base editing at the *HBG* promoter

ABE and CBE are known to create Cas-dependent DNA off-target and transient Cas-independent RNA off-target at low levels (*Anzalone et al., 2020*). It has been reported that the Cas-independent DNA off-target is very low and undetectable (*Anzalone et al., 2020*). We used Cas-OFFinder tool to predict the Cas-dependent DNA off-target for the novel gRNA (gRNA-11). The identified target regions were deep sequenced by NGS. Despite the higher on-target efficiency, off-target editing was not observed at the top target sites (*Figure 6a*). Next, we performed transcriptome-wide RNA

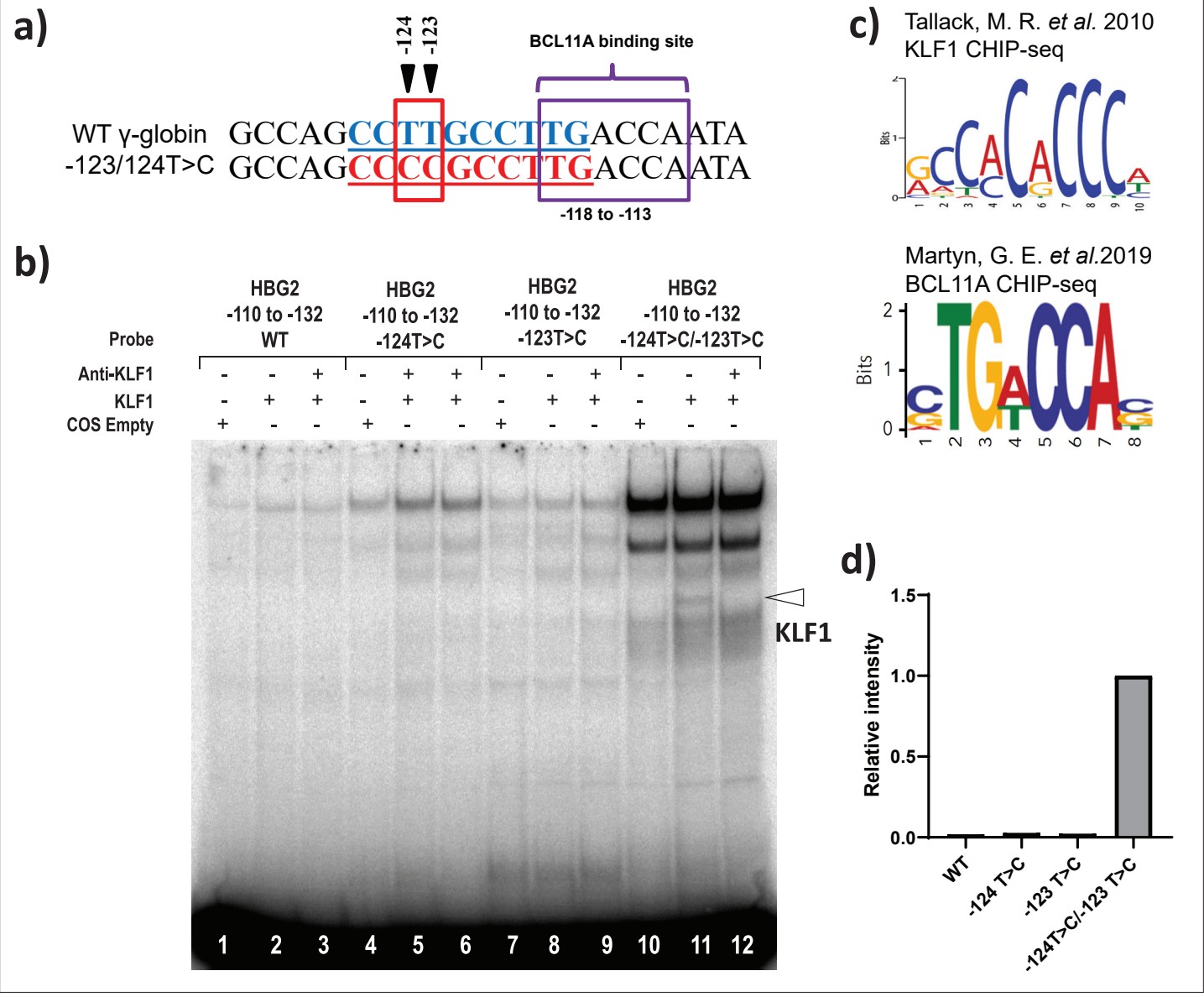

**Figure 5.** KLF1 binds to the −123T > C and -124T > C region of the *HBG* proximal promoter in vitro. (**a**) Introduction of T-to-C mutation at −123 and −124 of the *HBG* promoter (−132 to −110 bp) creates the de novo binding site for the KLF1, the wild type and novel KLF binding motif is highlighted in blue and red, respectively. (**b**) In vivo binding motifs of transcription factors KLF1 and BCL11A as determined by ChIP-Seq as previously reported. (**c**) Electrophoretic mobility shift assay (EMSA) showing KLF1 binding to −123T > C/−124T > C probe but failing to bind to −124T > C probe, −123T > C probe and WT probe with the −123T/−124T region of the *HBG* promoter in vitro. Lanes 1, 4,7, and 10 contain nuclear extracts from COS cells transfected with a pcDNA3 empty vector. Lanes 2–3, 5–6, 8–9, and 11–12 contain nuclear extracts from COS cells overexpressing KLF1. Binding of KLF1 to the −123T > C/−124T > C HPFH mutant probe can be observed in lane 11, with a super shift of KLF1 in the presence of anti-KLF1 antibody in lane 12. (**d**) Quantification of relative intensity of bands (KLF1 binding to the probe) from the EMSA using Image Lab 6.0.1 (Bio-Rad) software.

The online version of this article includes the following source data and figure supplement(s) for figure 5:

**Source data 1.** Electrophoretic mobility shift assay (EMSA) showing KLF1 binding to −123T > C/−124T > C probe but failing to bind to −124T > C probe, −123T > C probe, and wild type (WT) probe with the −123T/−124T region of the *HBG* promoter in vitro.

**Figure supplement 1.** Recruitment of KLF1 to site at −123 bp of *HBG* proximal promoter were analyzed by electrophoretic mobility shift assay (EMSA) and ChIP-qPCR.

**Figure supplement 1—source data 1.** Electrophoretic mobility shift assay (EMSA) showing the binding of KLF1 to the −123T > C/−124T > C probe but fails to bind to a wild type (WT) probe containing the −123/−124 region of the *HBG* promoter in vitro.

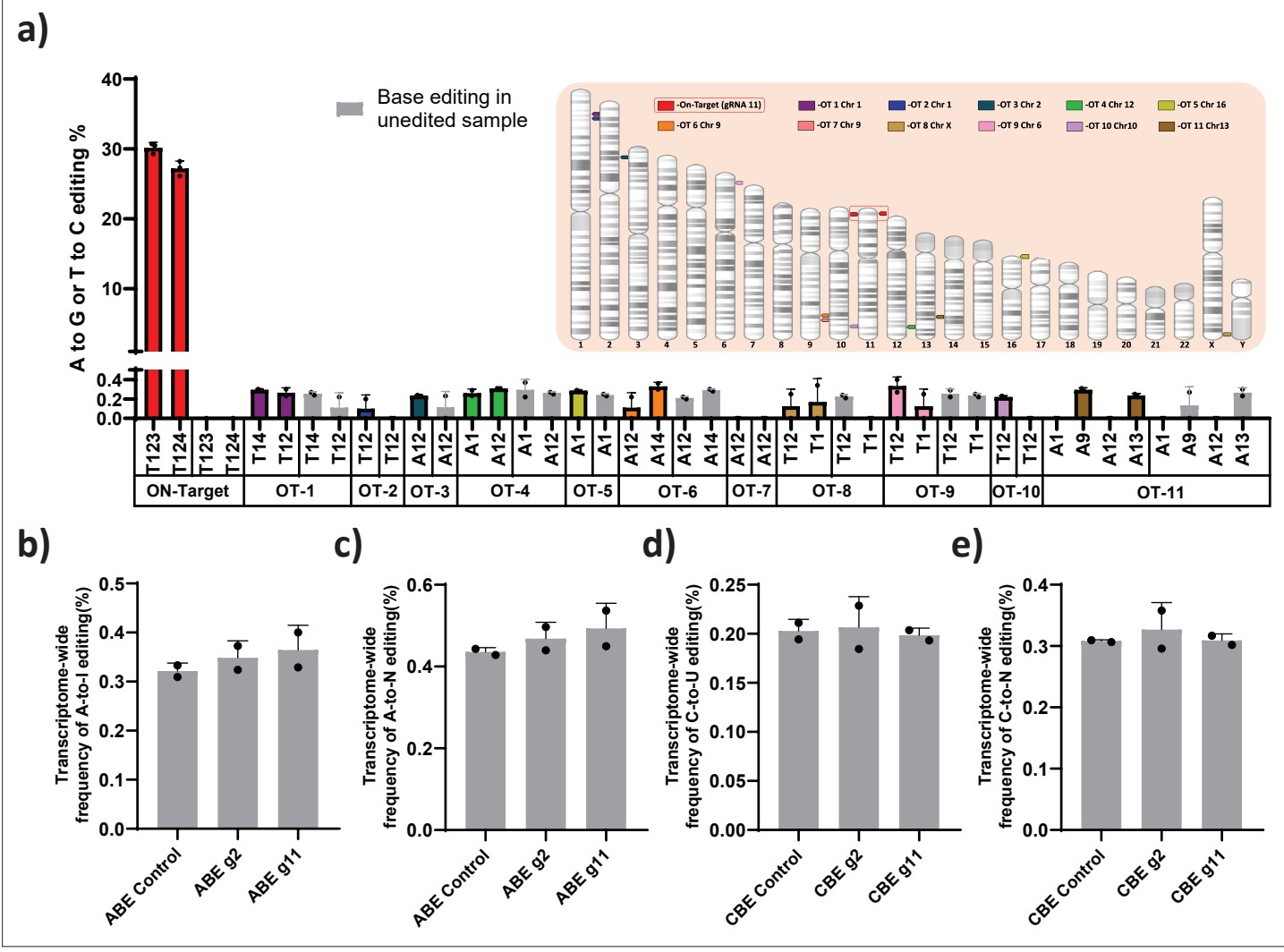

**Figure 6.** Evaluation of Cas-dependent DNA off-target and Cas-independent RNA off-target editing by adenine base editor. (**a**) Base conversions at the top 11 Cas-dependent DNA off-target sites in adenine base editor (ABE) 7.10 stable edited with guide RNA (gRNA)-11, along with the on-target events. The positions of the off-target and on-target loci are represented in their respective chromosome. The frequency of transcriptome-wide cellular levels of A- to- I (**b**), A- to- N (**c**), C- to -U (**d**), and C- to- N (**e**) RNA editing in BE3 stables (CBE), ABE 7.10 stables (ABE), BE3 stables edited with gRNAs-2 or -11, and ABE edited with gRNAs-2 or -11 are represented. The data are mean ± SD of two technical replicates.

The online version of this article includes the following figure supplement(s) for figure 6:

**Figure supplement 1.** Expression profile of *HBG* regulators after base editing.

sequencing on ABE and CBE stables with or without gRNA-2 and -11 to check whether base editing induced major spurious RNA deamination. The distribution frequency of A-to- I (in ABE) or C-to-U (in CBE) conversion across the base edited samples was very similar to that of the parental stable cell line (*Figure 6b–e*). To further verify that editing the gamma-globin promoter is not affecting the expression of other genes involved in globin regulation, we performed the differential analysis on these samples for the specific genes involved in globin regulation. We observed that there is no significant difference between the edited and control cells except for both gamma- and delta-globin genes (*Figure 6—figure supplement 1*).

Overall, these results support that the ABE and CBE are useful in creating specific point mutations in the homologous *HBG1* and *HBG2* promoters, leading to a potential increase in the number of HbF positive cells and overall HbF production with a significant reduction in the larger deletion frequency.

## Discussion

During normal globin switching, interactions of *cis*-acting elements with several different transcription factors lead to the silencing of fetal globin and in turn the activation of beta-globin (*Ikuta et al., 1996*). To obtain insights into the regulation of gamma-globin gene expression, we have used two complementary base editing approaches to screen the *HBG* promoter at single nucleotide resolution. This approach allowed us to identify several novel nucleotide substitutions in the *HBG* promoter that elevate HbF levels by altering the binding site for transcriptional activators or repressors.

Current approaches to studying fetal globin regulation by programmable nucleases often result in the deletion of the *HBG2* gene due to the introduction of DSBs in both *HBG* promoters (*Traxler et al., 2016*). The elimination of the 4.9 kb intergenic region (including the *HBG2* gene) appears to allow the locus control region (LCR) to directly interact with the *HBG1* promoter and drive its expression (*Métais et al., 2019*). It can be challenging to determine the exact role of different HPFH mutations on individual gamma-globin expression because mutations can occur in either or both *HBG2* and *HBG1* promoters. Further, the CRISPR-Cas9-based editing produces different combination of indel at the target sites which makes it difficult to pinpoint the precise mutations involved in the gene regulation. A base editing strategy converts target bases in the editing window without the generation of DSBs and hence largely avoids splicing of the *HBG* locus. Using this strategy, we targeted regions in both *HBG1 and HBG2* promoters and were able to efficiently edit sites in the promoters with fewer or no large deletions, which gave us the opportunity to evaluate gamma-globin expression from two active promoters.

We did observe a small percentage of 4.9 kb deletions even with base editors that use a nickase variant of the CRISPR/Cas9 system in our study. The larger deletions may be mainly a result of simultaneous CRISPR-Cas9-induced DSBs or by paired nickase-mediated two single-strand breaks (SSBs) on opposite DNA strands of the *HBG1* and *HBG2* gene (*Ran et al., 2013a*). Interestingly, recent studies have shown the possibility of adjacent SSB on the same DNA strand leading to the formation of genomic deletions in plants. The deletion frequency depends upon the initial release of the single-stranded fragment between the two SSBs (*Schiml et al., 2016*). Further reports suggest the conversion of the persistent nick into DSBs by the replication fork. The R-loop primed replication fork encounters the single-strand nick site in DNA template and collapses to produce a DSB (*Kuzminov, 2001*; *Wimberly et al., 2013*). Based on these findings, we predict the concurrent introduction of SSB by base editors at the editing site of the *HBG1* and *HBG2* promoter might generate some 4.9 kb larger deletions, though we observed very few.

Several different HPFH point mutations have been reported in the *HBG* promoters; and the effect of these mutations on gamma-globin expression in the native cellular environment has been deciphered for this limited set of mutations (*Bauer and Orkin, 2012*; *Liu et al., 2018*; *Martyn et al., 2019*; *Wienert et al., 2017*; *Wienert et al., 2015*). Our findings are in agreement with previous reports that the point mutations in three different regions of the *HBG* promoters centered around positions −198, −175, and −115 mimic the HPFH-associated point mutations affecting essential regulators of HbF expression (*Liu et al., 2018*; *Martyn et al., 2019*; *Martyn et al., 2018*; *Stoming et al., 1989*; *Wienert et al., 2017*; *Wienert et al., 2015*). Among the known HPFH point mutations, base conversion within the −115 cluster (from −110 to −116) showed the highest increase in promoter activity, confirming previous studies (*Fucharoen et al., 1990*; *Gilman et al., 1988*; *Zertal-Zidani et al., 1999*; *Motum et al., 1994*). CBE-mediated base conversion (C- to- T) at positions −114 and −115 resulted in a significantly greater induction of HbF than the multiple A-to-G nucleotide substitutions at −110, −112, −113, and −116 positions made by ABE. Recently, it has been shown that the major HbF repressor BCL11A directly binds to the core TGACC motif located at − 114 to −118 (*Liu et al., 2018*; *Martyn et al., 2018*). Naturally occurring HPFH mutations at −117G > A, −114C > A, −114C > T, −114C > G, and Δ13bp disrupts binding of BCL11A to the promoter (*Martyn et al., 2018*). The −113A > G HPFH mutation within the −115 cluster creates a binding site for the master erythroid regulator GATA1 without disrupting the binding of BCL11A (*Martyn et al., 2019*). Our results are consistent with these previous reports showing that disruption of the core binding region of BCL11A and the creation of a de novo binding sites for GATA1 results in the elevation of fetal globin in wild type HUDEP-2 cells (*Wienert et al., 2017*). ABE-mediated T-to-C substitution at position −198 of the *HBG* gene promoter has previously been shown to be associated with British HPFH and substantially elevates expression of HbF by creating a de novo binding site for the erythroid gene activator KLF1 (*Tate et al., 1986*;

*Wienert et al., 2017*). Another known HPFH mutation (–175T > C) has been shown to promote enhancer looping to the *HBG* promoter through recruitment of the activator TAL1 (*Wienert et al., 2015*). Further, increased editing efficiency at the –175T > C position with the hyperactive variants of ABE (ABE8e) resulted in the highest induction of HbF synthesis in human erythroid cells.

In this study, we have identified several new point mutations in the *HBG* promoter associated with high HbF levels. *HBG* promoter base editing by ABE-mediated conversion (A- to-G) revealed multiple potential HbF regulatory regions compared to CBE since the targeted region had more ABE-compatible gRNAs than CBE. In addition to the known mutations, we have identified novel substitutions at –69 (C -to- T), –70 (C- to- T), –122 (G -to -A), –123 (T -to- C), and –124 (T -to -C) of the *HBG* promoters as potential new regulatory mutations that can elevate gamma-globin expression. The levels of gamma-globin expression resulting from these mutations were very similar to those of well-characterized, naturally occurring HPFH mutations. Our study has predicted that nucleotide substitutions at –123T > C and –124T > C positions of the *HBG* promoter might result in reactivation of gamma-globin expression through the creation of a binding site for KLF-1, which was then confirmed by EMSA. This result, together with the observation that a de novo KLF1 site formed by the –198T-to-C mutation can upregulate fetal globin (*Tate et al., 1986*; *Wienert et al., 2017*) raises the possibility that introduction of a KLF1 binding site anywhere around the *HBG* promoter could potentially upregulate *HBG* gene expression. In contrast to our finding in EMSA, we observed only a very weak signal for the binding of KLF1 at the edited site of the *HBG* promoter by ChIP. Thus, our hypothesis, primarily on the basis of observing in vitro binding of KLF1 in EMSAs, is that the –123 and –124 mutations create a new KLF1 binding site, that is relatively weak and difficult to detect using ChIP but other hypotheses are possible. For instance, it could create a binding site for another activator. The relative proximity of this site to the BCL11A site, that begins around –117, suggests it may also directly or indirectly affect BCL11A binding. Further work needs to be done to assess these possibilities.

The current screening approaches that we used to identify the regulatory element in the proximal promoter of *HBG* is limited by several technical issues. The availability of NGG PAM sequences in the target region confines the resolution of the screening approach. The editing efficiency for ABE7.10 RA or BE3RA-FNLS is not uniform across the target regions (*Koblan et al., 2018*). The effect of transverse mutation in the target region on gene regulation is not possible as the current base editors are mainly involved in the installation of transition mutations (*Gaudelli et al., 2017*; *Komor et al., 2016*). The bystander mutation introduced by the base editors at the target regions makes it difficult to identify functional regulatory single nucleotides responsible for the gamma-globin regulation. These limitations can be overcome by the use of several different strategies including the use of alternative base editor variants that recognize the non-canonical PAM site (*Richter et al., 2020*). In addition, recently developed hyperactive variant of base editors will improve the increasing editing efficiency at the target site with the broader editing window (*Richter et al., 2020*). The scope of this study can be further increased by the dual ABE and CBE that can mediate both conversions (A-to-G and C-to-T) simultaneously, and also by prime editing approach which can widen the range of precise conversions in the desired region (*Anzalone et al., 2019*; *Zhang et al., 2020*).

The translational potential of genome edited HSPCs depends on long-term engraftment and repopulation ability. However, genotoxicity and cytotoxicity that can arise as a result of DSBs generated by programmable nucleases can be a limiting factor (*Cullot et al., 2019*; *Yu et al., 2016*). A previous study in nonhuman primates observed that the *HBG* promoter editing by Cas9 resulted in *HBG2* deletion with up to 27% frequency and that cells with this deletion were under-represented after engraftment (*Humbert et al., 2019*). Base editing at the target sites of *HBG1* and *HBG2* promoter by ABE and CBE does not result in high frequency of large deletions in the intergenic region as seen with Cas9 and only showed low levels of indel formation. ABEs have an inherent advantage over CBEs as they generate desired edits (A:T to G:C) with high fidelity, whereas the latter generate unanticipated edits. In corroboration with existing findings, our results also suggest that ABE is a better base editor than CBE with respect to purity of base conversion and indel formation (*Lee et al., 2018*). Moreover, preliminary results from our study suggest that the base editing of the HSPCs by ABE8e variant with the novel site (by gRNA-11) elevated HbF to therapeutic levels in erythroid progeny. Further, our study did not observe any significant DNA and RNA off-target in the ABE and CBE edited cells. Our proof of principle study validated the various gRNAs that can elevate the HbF levels to therapeutic

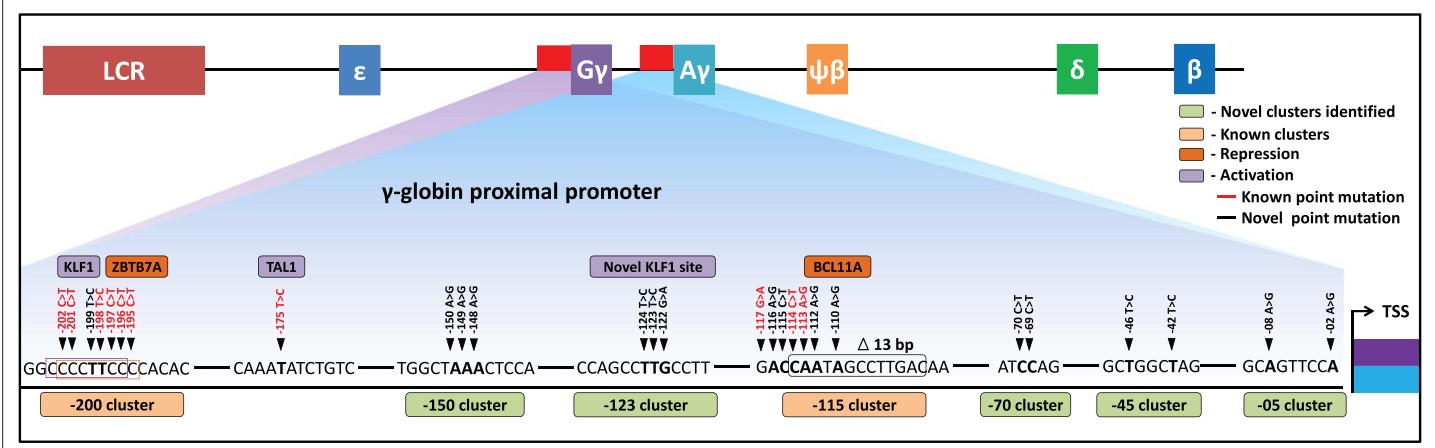

**Figure 7.** Schematic representation of known and identified point mutations in *HBG* promoter region that elevates fetal hemoglobin (HbF): The proximal promoter region of *HBG2* and *HBG1* is represented from transcription start site (TSS) till –205 bases.
Novel clusters identified from this study are highlighted in Sage (five clusters), and known clusters are highlighted in Melon (two clusters). Among these clusters, known base conversions are represented in black and identified hereditary persistence of fetal hemoglobin (HPFH)-like mutations are represented in red text. The novel base conversions from our study are represented in bold font. Transcriptional activators (lavender) and repressors (orange) that bind to the known clusters are also depicted in the figure.

levels laying the groundwork for potential clinical applications. This approach could address a range of beta-globin disorders avoiding the need to develop specific therapeutic products for each of them.

In summary, we have demonstrated that CRISPR base editing can be utilized to drive the expression of HbF to therapeutically relevant levels in an erythroid progenitor cell line and in HSPCs. After screening every gRNAs within the 320 bp region of the *HBG* promoter, we identified nine gRNAs that, when paired with the appropriate base editor, can introduce HPFH-like mutations without the generation of indels. We identified five novel regulatory regions for *HBG1* and *HBG2* that are required for the silencing of gamma-globin in adult erythroid cells shedding light on the molecular mechanisms behind hemoglobin switching (*Figure 7*). Our work is an exemplification of base editors in mapping gene regulatory elements in highly homologous locus and we hope base editing strategy will be among the pre-eminent therapeutic strategies for monogenetic disorders like beta-hemoglobinopathies in the future.

## Materials and methods
### Designing and cloning of the gRNA
The gRNAs for targeting the *HBG1* and *HBG2* promoter region were designed using SnapGene and Benchling. The gRNAs for CBE were designed using design-type 'gRNAs for base editing' in the Benchling tool; from the 43 hits, we selected 32 non-overlapping gRNAs. The gRNAs for ABE were designed manually using SnapGene software. The forward oligonucleotide consists of the gRNA sequence without PAM (20 bp) and 'CACCG' overhang at the 5' end, while the reverse oligonucleotide consists of reverse complement of gRNA without PAM (20 bp), 'AAAC' overhang at the 5' end and a 'C' added at 3' end. The synthetic complementary oligonucleotides listed in *Supplementary file 1* were annealed (*Ran et al., 2013b*; *Shalem et al., 2014*) and cloned into *BsmBI* digested pLKO5.sgRNA.EFS.GFP/RFP vector (gift from Benjamin Ebert, Addgene #57822/#57823) (*Heckl et al., 2014*). The oligo-annealed products were diluted 1:200-fold, from which 6 μl was taken along with 50 ng of vector backbone and ligation reaction was set up as per the manufacturer's instruction from NEB. The ligated product was transformed into DH10B competent cells and plated in LB agar containing 100 μg/ml of ampicillin for selection (*Sambrook and Russell, 2006*). Three colonies were picked from the plate and inoculated in LB for colony PCR. Colony PCR was carried out using GoTaq Hot Start Polymerase premix (Promega) and 1 μl each of forward and reverse sequencing primers (10 picomoles) (*Supplementary file 2*) along with 1 μl of processed cells in a thermocycler (Applied Biosystems Veriti). The cyclic conditions were as follows: initial denaturation at 95°C for 10 min, 35

cycles of 95°C for 30 s, 55°C for 30 s, 72°C for 45 s, followed by a final extension at 72°C for 7 min. After confirming the expected amplification in 1% agarose gel, second round of PCR was carried out using the 20 ng of pre-cleaned product from the first round of PCR using BigDye Terminator v3.1 Cycle Sequencing Kit as per manufacturer's protocol and given for Sanger sequencing.

## Plasmid constructs

The plasmids used in this study, pLenti-FNLS-P2A-Puro (Addgene#110841-CBE) and pLenti-ABERA-P2A-Puro (Addgene#112675-ABE), were a gift from Lukas Dow (*Zafra et al., 2018*), and pMD2.G and psPAX2 (second-generation lentiviral packaging construct, Addgene #12259, 11260) were a gift from Didier Trono. The pLenti-ABE8e-puro vector was constructed by amplifying ABE8e from the TadA-8e V106W plasmid (a plasmid gifted from David Liu, Addgene#138495) (*Richter et al., 2020*) using the primers mentioned in *Supplementary file 2*. The amplified PCR product was then cloned into pLenti-ABERA-P2A-Puro backbone after digestion with BamH1 and Nhe1 by HIFI assembly (NEB). The gRNA sequence from lentiCRISPR V2 vector (a construct gifted from Feng Zhang, Addgene#52961) was removed by digestion with EcoR1 and Kpn1 enzyme (NEB). The digested plasmid was then re-ligated with NEB Ligase after a exonuclease treatment (NEB Exonuclease 1) to generate a lentiCRISPR V2.1 vector (*Sanjana et al., 2014*). The plasmids were isolated using NucleoBond Xtra Midi EF (Macherey-Nagel) according to the manufacturer's instruction.

## Cell culture

HUDEP-2 cell lines were cultured in StemSpan SFEM II (STEMCELL Technologies) supplemented with 50 ng/ml SCF (ImmunoTools), 3 U/ml EPO (Zyrop 4000 IU injection), 1× Pen-Strep (Gibco), 1 μM dexamethasone (Alfa Aesar), 1 μg/ml doxycycline (Sigma-Aldrich), and 1× L-glutamine 200 mM (Gibco) (*Kurita et al., 2013*). The cells were culture at 37°C with 5% $CO_2$ and were confirmed negative for mycoplasma (Universal Mycoplasma detection kit-ATCC). K562 cell line was cultured in RPMI (Roswell Park Memorial Institute media) (Hyclone) supplemented with 1× penicillin-streptomycin-glutamine (Gibco) and 10% fetal bovine serum (FBS) (Gibco). COS-7 cells and HEK 293T cells were cultured in Dulbecco's modified Eagle medium (DMEM, Gibco) supplemented with 10% (v/v) FBS and 1× Pen-Strep.

The left-over peripheral blood mononuclear cells (PBMNCs) were obtained from a healthy donor after infusion according to the clinical protocols approved by the Intuitional Review Boards of Christian Medical College, Vellore. The PBMNCs were purified by density gradient centrifugation (Lymphoprep Density Gradient Medium|STEMCELL Technologies) followed by RBC lysis. CD34+ cells were isolated from the purified PBMNCs by EasySep Human CD34 positive selection kit II (STEMCELL Technologies) and expanded in HSC expansion media as described earlier (*Genovese et al., 2014*). The isolated cells were analyzed for primitive cell surface markers (CD34+ CD133+ and CD90+) after 24 hr of expansion (*Genovese et al., 2014*).

## Lentivirus production

HEK293T cells ($1 \times 10^6$) were cultured in 10 cm cell culture dish (Corning). Around 80% confluency, 2.5 μg of pMD2.G (envelope plasmid), and 3.5 μg of psPAX2 (packaging plasmid) along with 4 μg (construct with gRNA) or 5 μg (construct with ABE/CBE/Cas9) of lentiviral vector were transfected using FuGENE-HD as per the manufacturer's protocol. The viral supernatants were separately collected at 48 and 72 hr; and concentrated using Lenti-X Concentrator (Takara). The concentrated pellet was resuspended in 200 μl of 1×PBS, and the aliquots were stored at –80°C.

## Lentiviral transduction

The desired lentivirus (100 μl aliquot) along with 6 μg/ml polybrene (Sigma-Aldrich), and 1% HEPES 1 M buffer (Gibco) were added to HUDEP-2 or K562 cells (0.5 million cells in one well of a six-well plate) and spinfected at 800 g for 30 min at room temperature. The cells were incubated for 48 hr with lentivirus at 37°C and then incubated in fresh medium. For the stable cell line generation, the cells transduced with pLenti-FNLS-P2A-puro or pLenti-ABERA-P2A-puro or pLenti-ABE8e-puro or lentiCRISPR V2.1viral vector were then treated with 1 μg/ml puromycin (Gibco) for 10 days. In case of gRNA transduction with pLKO5.sgRNA.EFS.GFP/RFP vector, the transduced cells were analyzed by FACS for GFP/RFP expression.

## In vitro transcription

The template for in vitro transcription (IVT) was prepared by linearizing ABE8e plasmid (Addgene#138495) with Pme1 (NEB) and purified using PCI (phenol-chloroform-isoamyl alcohol). IVT was carried out using T7 mScript Standard mRNA Production System (CELLSCRIPT) components by previously described method with full substitution of pseudouridine-5'-triphosphate (Jena Bioscience) for uridine (*Mahalingam et al., 2022*). The purified mRNA was stored as aliquots (5 µg/vial) in –80°C.

## Electroporation of CD34+ cells

CD34+ cells were expanded in HSC expansion media for 48 hr. Around 1 million of CD34+ cells were pelleted then resuspended in 19 µl MaxCyte buffer (Hyclone) along with 5 µg ABE8e mRNA (5 µl) and 100 pmole desired gRNA (1 µl) (Synthego) (target information in *Supplementary files 1 and 2*). The resuspended cells were loaded into one well of OC25 × 3 Maxcyte cuvette and electroporated with program 'HSC-3'. After electroporation, the content was transferred to single well of 12-well plate (Corning) and allowed to recover for 20 min in the incubator (5% $CO_2$, 37°C). To the recovered cells, 1 ml of HSC expansion media was added and then expanded for 48 hr before performing any further experiments.

## Erythroid differentiation

For the erythroid differentiation of HUDEP-2 cells, we followed previously established protocol with slight modification (*Trakarnsanga et al., 2017*). After 8 days of expansion, around 1 million of edited cells were seeded in 65 mm cell culture dish (Eppendorf) with 5 ml of differentiation media consisting of IMDM glutamax (Gibco), 3% AB serum (MP Biomedicals), 2% FBS, 0.1% insulin solution human (Sigma-Aldrich), 3 U/ml Heparin sodium salt (MP Biomedicals), 200 µg/ml Holo Transferrin (BBI Solutions), 3 U/ml EPO, 10 µg/ml SCF, 1 ng/ml IL3 (Immuno Tools), 1× Pen-Strep, and 1 µg/ml doxycycline. Erythroid differentiation was carried out in 10 cm dish with regular media change (on days 3 and 6) up to the end of differentiation (for 9 days). On day 6, these cells were cultured in erythroid differentiation medium with 500 µg/ml of holotransferrin and devoid of doxycycline.

For erythroid differentiation of CD34+ cells, HSPCs were cultured in a three-phase liquid culture system and subjected to enucleation analysis as previously described (*Psatha et al., 2018*). The erythroid differentiation pattern was evaluated in the erythroblast obtained from HUDEP-2 cells (on day 9) and CD34+ cells (on day 21) by FACS analysis of CD235a and CD71 markers.

## Analysis of base editing efficiency

Genomic DNA was isolated from the edited samples using DNA isolation kit (NucleoSpin Blood – Macherey-Nagel). For Sanger sequencing, the targets were PCR amplified using GoTaq Hot Start Polymerase premix (Promega), the primers used are listed in *Supplementary files 1 and 2*. For NGS, the targets were PCR amplified (the primers listed in *Supplementary files 1 and 2*) using GXL premix (Takara Bio) and sequenced using MiSeq System (Illumina). The library preparation and sequencing was carried out as per previously described protocol (*Corn, 2017*). The Fastq files obtained were analyzed for base editing using CRISPResso-2 (*Clement et al., 2019*). The data obtained from Sanger sequencing were used to analyze indels and the base editing efficiency by tools like Inference of CRISPR Edits (ICE) (Synthego) and EditR, respectively (*Hsiau et al., 2018*; *Kluesner et al., 2018*).

For characterization of editing in individual *HBG1* and *HBG2* promoter, NGS 4F and NGS 2R primers (*Supplementary files 1 and 2*) were used to amplify *HBG* promoter. After NGS , the Fastq file obtained were aligned to *HBG1* and *HBG2* sequence using Bowtie2 based on nucleotide variation at −307, −317, and −324. The aligned reads were visualized using IGV (*Robinson, 2012*; *Langmead and Salzberg, 2013*) and the editing efficiency was computed individually for both the genes ((edited reads/total reads) ×100).

## Real-time PCR

Total RNA from the edited cells were isolated using the NucleoSpin RNA kit (Macherey-Nagel) and reverse transcribed using cDNA Synthesis Kit (iScript Bio-Rad). The relative expression (ΔΔCT) of *HBB*, *HBA*, and *HBG* genes was determined using the respective primers (*Supplementary file 2*) by qRT-PCR using SsoFast EvaGreen Supermixes (Bio-Rad) in QuantStudio 6 Flex Real-Time PCR System (Applied Biosystems). The qRT-PCR mixture (10 µl) contains 1 µl each of respective forward and reverse primer

(5 μM), 5 µl of SYBR green master mix, 2 µl of H$_2$O, and 1 µl of 5-fold diluted cDNA template. *GAPDH* was used as an internal control gene to normalize the data for ΔΔCT (relative expression analysis). The cycling condition was performed as per the manufacturer's protocol (Bio-Rad). A dissociation curve analysis was carried out to ensure there is no unspecific amplification.

The VCN was assessed in genomic DNA isolated from the transduced samples using qRT-PCR as previously described with a few modification (*Barczak et al., 2015*). Primers targeting U6 promoter (for gRNA integration), *Cas9* gene (for Cas9 and base editors' integration), and WPRE (for gRNA and Cas9 variant integration together) were used. Exon 2 of *HBB* gene was used as a single copy gene-specific reference. The primers used are listed in *Supplementary file 2*. pLKO5.sgRNA. EFS.GFP (Plasmid #57822, Addgene) and an inhouse plasmid carrying *HBB* CDS (details not provided) were used as standards.

## HbF intracellular staining

To evaluate the frequency of HbF positive cells, the cells were fixed, permeabilized, and intracellular staining was performed using Fetal Hemoglobin Monoclonal Antibody (HBF-1), APC (Invitrogen) as previously described (*Canver et al., 2015*). The stained cells were analyzed by FACS (BD FACSAria III Cell Sorter or CytoFLEX LX Flow Cytometer – BC) to measure the number of HbF positive cells.

## Hemoglobin detection by HPLC

The differentiated cells were collected and washed with 1× PBS and resuspended in 1100 µl cold ddH$_2$O. The cells were sonicated for 30 s with 50% Amp in ice and centrifuged at 14,000 rpm for 15 min at 4°C. The supernatant (1000 µl) was analyzed for hemoglobin variants by VARIANT II Hemoglobin Testing System (Bio-Rad). The hemoglobin percentages were calculated by the Bio-Rad's Clinical Data Management (CDMTM) Software. Reverse phase HPLC (Shimadzu Corporation-Phenomenex) (*Loucari et al., 2018*) was performed in remaining 100 µl of the supernatant for the analysis of individual globin chains expression . The ratio of gamma (A and G gamma)/beta-like (gamma, beta, and delta) globin was calculated and represented in percentage.

## Validation of 4.9 kb large deletion

To quantify the large deletion in *HBG* promoter region, qPCR was carried as previously reported (*Li et al., 2018*) (using primers from *Supplementary file 2*). To verify the effect of larger deletion on gamma-globin expression, the globin chain analysis was carried out using RP HPLC in the differentiated erythroid cells. The A gamma- and G gamma-globin chain percentage obtained from each sample were normalized with control.

## COS-7 cell transfections and nuclear extraction

COS-7 cells were transfected with 5 µg of mammalian expression plasmids pcDNA3-empty (Invitrogen) or pSG5/mEKLF-Mouse (*Miller and Bieker, 1993*) using FuGENE 6 (Promega) in 10 cm culture dish, according to the manufacturer's instructions. Transfected cells were incubated at 37°C for 48 hr before harvest. Nuclear extractions were performed as previously described (*Andrews and Faller, 1991*).

## Electrophoretic mobility shift assay

Oligonucleotides used in radiolabelled probes are listed in *Supplementary file 2*. The sense strand for each probe was labelled with P-32 from γ-$^{32}$P ATP (Perkin Elmer) using T4 PNK (NEB), before annealing the antisense strand by slow cooling from 100°C to room temperature. The annealed probes were purified using quick spin columns for radiolabelled DNA purification (Roche). Plasmids were overexpressed and harvested from COS-7 cells, and 'empty' extract without the target protein was used to aid identification of background bands caused by endogenous protein binding. Antibody for KLF1 was used as indicated to identify the protein on the gel (*Crossley et al., 1996*). Complexed samples were loaded on 6% native polyacrylamide gel in TBE buffer (45 mM Tris, 45 mM boric acid, 1 mM EDTA). Electrophoresis was performed at 4°C and 250 V for 1 hr and 40 min, and then vacuum dried before exposing a FUJIFILM BAS Cassette2 phosphor screen overnight. Imaging was performed on a GE Typhoon FLA 9500 fluorescent image analyzer.

## ChIP qPCR

Each immunoprecipitation was performed using $5 \times 10^7$ cells of wild type and edited HUDEP-2 cells before differentiation. Cells were cross-linked in 1% formaldehyde solution and incubated at room temperature for 10 min before the reaction was quenched by addition of glycine to a final concentration of 125 mM. Cross-linked cells were lysed and sonicated for 10 cycles of 30 s with 30 s intermissions at 4°C to obtain chromatin fragments of approximately 200–300 bp. Immunoprecipitations were performed using 100 µl of Dynabeads Protein G (ThermoFisher Scientific) complexed to 15 µg of KLF1 antibody (OriGene, #TA305808) or normal rabbit IgG (Cell Signaling Technology #2729S) at 4°C overnight. Magnetic beads were separated and washed thoroughly before elution and cross-linking was reversed by incubation at 65°C overnight. DNA was then purified and quantified within reference to whole cell extract on a ViiA 7 Real-Time PCR System using SYBR green reagents and the ΔΔCt method for specific targets (*Supplementary file 2*).

## RNA sequencing and analysis

Total RNA extracted using NucleoSpin RNA kit (Macherey-Nagel) was quality assessed by Agilent 2100 Bioanalyzer (Agilent Technologies). From 1 µg of total RNA polyadenylated transcripts was purified using oligo-dT beads (TruSeq RNA Sample Preparation Kit, Illumina). Fragmentation was carried out in the presence of divalent cation followed by reverse transcription using Superscript II Reverse Transcriptase kit (Life Technologies). Following cDNA purification by Ampure XP SPRI beads (Beckman Coulter) Illumina adapter ligation and amplification were carried out. The quantification and the quality were assessed by NanoDrop spectrophotometer (Thermo Scientific) and Bioanalyzer (Agilent Technologies), respectively. Libraries were sequenced by using Illumina NovoSeq 6000 platform as 150 bp paired-end reads. Fastq files were generated with bcl2fastq and then trimmed to remove low-quality bases, adapter seq, and unpaired sequence using TrimGalore. *Homo sapiens* genome assembly GRCh38 was used as a reference to align the trimmed reads. NFCore RNA Seq pipeline was used to resolve the expressed transcripts quantitatively and qualitatively (*Ewels et al., 2020*). The files are accessible through the GEO Series accession number GSE192801.

Transcriptome analysis was carried out in wild type HUDEP-2, ABE, and CBE stable cells with or without gRNA-2 and -11 in duplicate. The transcript was counted from the sorted bam files by the aligner mentioned above. Interactive Gene Expression Analysis Kit (iGEAK) RNA-seq v1.0 a R and JavaScript-based tool was used to normalize gene expression levels and perform differential expression analysis (*Choi and Ratner, 2019*).

## Off-target analysis

Cas-OFFinder was used to find the Cas-dependent DNA off-target, up to three mismatches were allowed in selecting targets (target information in *Supplementary file 3*). The targets were amplified and sequenced using Illumina MiSeq platform (using primers mentioned in *Supplementary file 2*). CRISPResso2 was used to align the reads, only high-quality reads were used for this analysis (q = 30). REDItools v2 was used to calculate the transcriptome-wide A-to-I and C-to-U conversion in ABE and CBE edited samples. Except the respective nucleotide (A for ABE and C for CBE), all nucleotides were removed from the analysis. Read coverage and read quality criteria were followed as described earlier (*Koblan et al., 2021*). The frequency of A converted to I/N and C converted to U/N was calculated by dividing the total number of converted nucleotides by the respective nucleotides after filtering (A-to-(I or N)/A*100 or C-to-(U or N)/C*100). The experiment was carried out as two biological replicates.

## Statistical analysis

The statistical tests were performed using GraphPad Prism 8.1. Since all the data were normally distributed, unpaired two-sided t-test or one-way ANOVA was used as appropriate. In all the tests, $p < 0.05$ was considered statistically significant. Linear regression was carried out to find out if any correlation exists between two variables. Also, to find the relationship between the samples, Pearson correlation was performed. Principal component analysis (PCA) was performed using R statistical package.

## Acknowledgements

The research reported in this work was supported by NAHD grant: BT/PR17316/MED/31/326/2015 (Department of Biotechnology, New Delhi, India), EMR grant: EMR/2017/004363 (Science and Engineering Research Board [SERB], New Delhi, India), Indo-US GETin Fellowship_2018_066 (Indo-US Science & Technology Forum [IUSSTF]), and DBT grant: BT/PR38392/GET/119/301/2020. We sincerely acknowledge CSCR (a unit of inStem, CMC Campus, Vellore, India) for providing the startup funds. NSR and AG is supported by Senior Research Fellowship from Council of Scientific & Industrial Research India. VR is supported by Senior Research Fellowship DBT India. BW is supported by an Early Career Research Fellowship, and HWB and MC were supported by a grant from the National Health and Medical Research Council Australia. HWB is additionally supported by an Australian Government Research Training Program Scholarship. SM is supported by gene editing task force (DBT); Grant No# BT/PR25841/GET/119/162/2017. We thank Mrs Sumithra and Mr Neelagandan at the Department of Hematology, CMC, for help with HPLC variants; Keerthivasan. RC, IISER Mohali, and Ashis Kumar S, CSCR, for bioinformatics. Also, we like to acknowledge the CSCR core facility for supporting us with all the required instrumentations.

## Additional information

### Funding

| Funder | Grant reference number | Author |
| --- | --- | --- |
| Ministry of Science and Technology | BT/PR17316/MED/31/326/2015 | Kumarasamypet M Mohankumar |
| Science and Engineering Research Board | EMR/2017/004363 | Kumarasamypet Murugesan Mohankumar |
| Indo-US Science and Technology Forum | Indo-U.S. GETin Fellowship_2018_066 | Kumarasamypet Murugesan Mohankumar |
| Ministry of Science and Technology | BT/PR38392/GET/119/301/2020 | Kumarasamypet M Mohankumar |
| Council of Scientific and Industrial Research, India | Senior Research Fellow | Nithin Sam Ravi Anila George |
| Ministry of Science and Technology | Senior Research Fellow | Vignesh Rajendiran |
| National Health and Medical Research Council | Early Career Research Fellowship | Beeke Wienert |
| Ministry of Science and Technology | BT/PR25841/GET/119/162/2017 | Srujan Marepally |
| National Health and Medical Research Council | National Health and Medical Research Council (NHMRC) | Henry William Bell |
| National Health and Medical Research Council | Grant | Merlin Crossley |
| National Health and Medical Research Council | | Henry William Bell |

The funders had no role in study design, data collection and interpretation, or the decision to submit the work for publication.

### Author contributions

Nithin Sam Ravi, Data curation, Formal analysis, Investigation, Methodology, Resources, Software, Validation, Visualization, Writing – original draft, Writing – review and editing; Beeke Wienert, Resources, Visualization, Writing – review and editing; Stacia K Wyman, Methodology, Resources, Software; Henry William Bell, Jonathan T Vu, Aswin Anand Pai, Poonkuzhali Balasubramanian, Methodology, Resources; Anila George, Formal analysis, Writing – review and editing, Methodology; Gokulnath Mahalingam, Kirti Prasad, Bhanu Prasad Bandlamudi, Nivedhitha Devaraju, Vignesh

Rajendiran, Nazar Syedbasha, Methodology; Yukio Nakamura, Ryo Kurita, Resources; Muthuraman Narayanasamy, Writing – review and editing; Saravanabhavan Thangavel, Srujan Marepally, Resources, Writing – review and editing; Shaji R Velayudhan, Mark A DeWitt, Methodology, Resources, Writing – review and editing; Alok Srivastava, Project administration, Resources, Writing – review and editing; Merlin Crossley, Investigation, Methodology, Resources, Visualization, Writing – review and editing; Jacob E Corn, Methodology, Resources, Visualization, Writing – review and editing; Kumarasamypet M Mohankumar, Conceptualization, Data curation, Formal analysis, Funding acquisition, Investigation, Methodology, Project administration, Resources, Supervision, Validation, Visualization, Writing – original draft, Writing – review and editing

### Author ORCIDs
Nithin Sam Ravi http://orcid.org/0000-0002-8063-6935
Henry William Bell http://orcid.org/0000-0003-4677-8208
Anila George http://orcid.org/0000-0002-6016-976X
Jonathan T Vu http://orcid.org/0000-0002-4950-7967
Jacob E Corn http://orcid.org/0000-0002-7798-5309
Kumarasamypet M Mohankumar http://orcid.org/0000-0001-9407-1800

### Ethics
The left-over peripheral blood mononuclear cells (PBMNC) were obtained from a healthy donor after infusion according to the clinical protocols approved by the Institutional Review Boards of Christian Medical College, Vellore. IRB Min. No. 12309 (OTHER) dated 30. 10.2019.

### Decision letter and Author response
Decision letter https://doi.org/10.7554/eLife.65421.sa1
Author response https://doi.org/10.7554/eLife.65421.sa2

## Additional files

### Supplementary files
• Supplementary file 1. The guide RNAs (gRNAs) used in this study to screen the *HBG* promoter region and their respective primer for sequencing.
• Supplementary file 2. All the PCR, qRT-PCR primers and probes used in this study.
• Supplementary file 3. The targets analyzed for DNA off-target.
• Transparent reporting form

### Data availability
The transcriptome data have been deposited in GEO under accession code GSE192801 All the raw data from this study have been deposited in Dryad (https://doi.org/10.5061/dryad.bzkh1897h).

The following datasets were generated:

| Author(s) | Year | Dataset title | Dataset URL | Database and Identifier |
|---|---|---|---|---|
| Ravi N, Wyman SK, Mohankumar KM | 2022 | Identification of novel HPFH-like mutations by CRISPR base editing that elevate the expression of fetal hemoglobin | http://www.ncbi.nlm.nih.gov/geo/query/acc.cgi?acc=GSE192801 | NCBI Gene Expression Omnibus, GSE192801 |
| Mohankumar KM | 2022 | Data from: Identification of novel HPFH-like mutations by CRISPR base editing that elevate the expression of fetal hemoglobin | http://dx.doi.org/10.5061/dryad.bzkh1897h | Dryad Digital Repository, 10.5061/dryad.bzkh1897h |

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

# Appendix 1

## Appendix 1—key resources table

| Reagent type (species) or resource | Designation | Source or reference | Identifiers | Additional information |
|---|---|---|---|---|
| Genetic reagent (*Homo sapiens*) | *GRCh38* | GenBank | 883148 | |
| Strain, strain background (*Escherichia coli*) | DH10B | ECOS, Yeastern Biotech | CAT # FYE507-10VL | |
| Recombinant DNA reagent | pLKO5.sgRNA.EFS.GFP | Addgene | Addgene_57822; RRID:Addgene_57822 | |
| Recombinant DNA reagent | pLKO5.sgRNA.EFS.RFP | Addgene | Addgene_57823; RRID:Addgene_57823 | |
| Recombinant DNA reagent | pLenti-FNLS-P2A-Puro | Addgene | Addgene_110841; RRID:Addgene_110841 | |
| Recombinant DNA reagent | pLenti-ABERA-P2A-Puro | Addgene | Addgene_112675; RRID:Addgene_112675 | |
| Recombinant DNA reagent | pMD2.G | Addgene | Addgene_12259; RRID:Addgene_12259 | |
| Recombinant DNA reagent | psPAX2 | Addgene | Addgene_12260; RRID:Addgene_12260 | |
| Recombinant DNA reagent | TadA-8e V106W | Addgene | Addgene_138495; RRID:Addgene_138495 | |
| Recombinant DNA reagent | lentiCRISPR V2 | Addgene | Addgene_52961; RRID:Addgene_52961 | |
| Recombinant DNA reagent | lentiCRISPRV2.1 | This study | | Cas9 expressing lentiviral plasmid without gRNA scaffold |
| Recombinant DNA reagent | pLenti-ABE8e-P2A-Puro | This study | | ABE8e expressing lentiviral plasmid |
| Recombinant DNA reagent | pcDNA-3 | Invitrogen | | |
| Recombinant DNA reagent | pSG5/mKLF | *Miller and Bieker, 1993* | | |
| Antibody | PE Mouse Anti-Human CD34 (mouse monoclonal) | BD Pharmingen | CAT # 550761; RRID:AB_393871 | FACS (2 µl/test) |
| Antibody | APC Mouse Anti-Human CD133 (mouse monoclonal) | BD Pharmingen | CAT # 566596; RRID:AB_2744280 | FACS (2 µl/test) |
| Antibody | BV421 Mouse Anti-Human CD90 (mouse monoclonal) | BD Pharmingen | CAT # 562556; RRID:AB_2737651 | FACS (2 µl/test) |
| Antibody | PE-Cy7 Mouse Anti-Human CD235a (mouse monoclonal) | BD Pharmingen | CAT # 563666; RRID:AB_2738361 | FACS (2 µl/test) |
| Antibody | BV421 Mouse Anti-Human CD71 (mouse monoclonal) | BD Pharmingen | CAT # 562995; RRID:AB_2737939 | FACS (2 µl/test) |
| Antibody | PE Mouse Anti-Human CD235a (mouse monoclonal) | BD Pharmingen | CAT # 555570; RRID:AB_395949 | FACS (2 µl/test) |
| Antibody | FITC Mouse Anti-Human CD71 (mouse monoclonal) | BD Pharmingen | CAT # 555536; RRID:AB_395920 | FACS (2 µl/test) |
| Antibody | Fetal Hemoglobin Antibody, APC (mouse monoclonal) | Invitrogen | CAT # MHFH05; RRID:AB_10374595 | FACS (2 µl/test) |
| Antibody | Antibody for KLF1 (rabbit polyclonal) | *Crossley et al., 1996* | | EMSA (1:30 final dilution) |

*Appendix 1 Continued on next page*

*Appendix 1 Continued*

| Reagent type (species) or resource | Designation | Source or reference | Identifiers | Additional information |
|---|---|---|---|---|
| Antibody | Anti-KLF1 antibody (goat polyclonal) | OriGene | #TA305808 | ChiP (15 µg/IP) |
| Antibody | Normal rabbit IgG | Cell Signaling Technology | #2729S | ChiP (15 µg/IP) |
| Commercial assay or kit | NucRed Live 647 ReadyProbes Reagent | Invitrogen | CAT # R37106 | FACS (2 µl/test) |
| Commercial assay or kit | Zymoclean Gel DNA recovery kit | Zymo Research | CAT # D4001 | |
| Commercial assay or kit | NucleoBond Xtra Midi | MN | REF # 740410 | |
| Commercial assay or kit | NucleoSpin RNA | MN | REF # 740955 | |
| Commercial assay or kit | NucleoSpin Blood – DNA kit | MN | REF # 740951 | |
| Commercial assay or kit | EasySep Human CD34 Positive Selection Kit | STEMCELL Technologies | CAT # 17856 | |
| Commercial assay or kit | T7 mScript Standard mRNA Production System | Cell Script | C-MSC100625 | |
| Commercial assay or kit | Radiolabelled DNA column | Roche | G25DNA-RO | |
| Commercial assay or kit | iScript cDNA Synthesis Kit | Bio-Rad | CAT # 1708891 | |
| Commercial assay or kit | BigDye Terminator v3.1 Cycle Sequencing Kit | Applied Biosystem | CAT # 4337458 | |
| Commercial assay or kit | SsoFast EvaGreen Supermix | Bio-Rad | CAT # 172–5200 | |
| Commercial assay or kit | Universal Mycoplasma detection kit | ATCC | CAT # 30–1012K | |
| Chemical compound, drug | T4 Polynucleotide Kinase | NEB | CAT # M0201 | |
| Chemical compound, drug | T4 DNA Ligase | NEB | CAT # M0202 | |
| Chemical compound, drug | NEBuilder HiFi DNA Assembly Master Mix | NEB | CAT # E2621 | |
| Chemical compound, drug | BamHI-HF | NEB | CAT # R3136 | |
| Chemical compound, drug | NhEI-HF | NEB | CAT # R3131 | |
| Chemical compound, drug | BsmB1 | NEB | CAT # R0580 | |
| Chemical compound, drug | KpnI-HF | NEB | CAT # R3142 | |
| Chemical compound, drug | EcoRI-HF | NEB | CAT # R3101 | |
| Chemical compound, drug | Exonuclease I (*E. coli*) | NEB | CAT # M0293 | |
| Chemical compound, drug | Pme1 | NEB | CAT # R0560 | |
| Chemical compound, drug | GoTaq Green Master Mix | Promega | CAT # M712B | |
| Chemical compound, drug | PrimeSTAR GXL Premix | Takara Bio | CAT # R051A | |
| Chemical compound, drug | DynaBeads PG | Invitrogen | 10003D | |

*Appendix 1 Continued on next page*

*Appendix 1 Continued*

| Reagent type (species) or resource | Designation | Source or reference | Identifiers | Additional information |
|---|---|---|---|---|
| Chemical compound, drug | Formaldehyde | Sigma-Aldrich | F8775 | 1% v/v final concentration |
| Chemical compound, drug | Glycine | Ajax Finechem | AJA1083 | 125 mM final concentration |
| Chemical compound, drug | γ-[Kuzminov, 2001]P ATP | Perkin-Elmer | BLU502A250UC | 1 µl/15 pmol probe |
| Chemical compound, drug | Insulin | Sigma-Aldrich | CAT # 11061-68-0 | |
| Chemical compound, drug | Heparin | MP Biomedicals | CAT # 9041-08-1 | |
| Chemical compound, drug | Holotransferrin | BBI Solutions | #SKU T101-5 | |
| Chemical compound, drug | SCF | Immuno Tools | CAT # 11343325 | |
| Chemical compound, drug | EPO | Zydus Nephrosciences | Zyrop 4000 IU Injection | |
| Chemical compound, drug | IL6 | Immuno Tools | CAT # 11340066 | |
| Chemical compound, drug | IL3 | Immuno Tools | CAT # 11340035 | |
| Chemical compound, drug | FLT3 | Immuno Tools | CAT # 11343305 | |
| Chemical compound, drug | TPO | Immuno Tools | CAT # 11344863 | |
| Chemical compound, drug | Hydrocortisone | MP Biomedicals | CAT # 2930949 | |
| Chemical compound, drug | AB Serum | MP Biomedicals | CAT # 101996 | |
| Chemical compound, drug | Penstrep | Gibco | CAT # 15140122 | |
| Chemical compound, drug | Dexamethasone | Alfa Aesar | CAS# 1177-87-3 | |
| Chemical compound, drug | Doxycycline | Sigma-Aldrich | CAS# 24390-14-5 | |
| Chemical compound, drug | Glutamine | Gibco | CAT # 25030081 | |
| Chemical compound, drug | FBS | Gibco | CAT # 10270106 | |
| Chemical compound, drug | PBS | Hyclone | CAT # SH30256.02 | |
| Chemical compound, drug | Polybrene | Sigma-Aldrich | CAS # 28728-55-4 | |
| Chemical compound, drug | Hepes | Gibco | CAT # 15630080 | |
| Chemical compound, drug | Puromycin | Gibco | CAT # A1113803 | |
| Chemical compound, drug | Pseudouridine | Jena Bioscience | CAT # NU-1139 | |
| Chemical compound, drug | Lymphoprep | STEMCELL Technologies | CAT # 07851 | |
| Chemical compound, drug | StemSpan SFEM-II | STEMCELL Technologies | CAT # 09655 | |
| Chemical compound, drug | DMEM | Hyclone | CAT # SH30243.01 | |

*Appendix 1 Continued on next page*

*Appendix 1 Continued*

| Reagent type (species) or resource | Designation | Source or reference | Identifiers | Additional information |
|---|---|---|---|---|
| Chemical compound, drug | RPMI | Hyclone | CAT # SH30027.01 | |
| Chemical compound, drug | IMDM-Glutamax | Gibco | CAT # 31980030 | |
| Chemical compound, drug | Fugene HD | Promega Corporation | CAT # E2312 | |
| Chemical compound, drug | Lenti-X concentrator | Takara | CAT # 631232 | |
| Chemical compound, drug | Maxcyte buffer | Hyclone | CAT # EPB1 | |
| Chemical compound, drug | LB Agar | HIMEDIA | M1151 | |
| Chemical compound, drug | LB Broth | HIMEDIA | M1245 | |
| Chemical compound, drug | Ampicillin Sodium Salt | SRL | 61,314 | |
| Chemical compound, drug | Triton X-100 | Fisher Scientific | CAS #:9002931 | |
| Chemical compound, drug | Glutaraldehyde | MP Biomedicals | CAT # 198,595 | |
| Biological sample | PBMNCs | CMC | IRB Min. No. 10,549 (others) dated 15/02/2017 | |
| Cell line (*Homo sapiens*) | HEK 293T | ATCC | | |
| Cell line (*Homo sapiens*) | HUDEP-2 | Cell Engineering Division, RIKEN BioResource Center | | |
| Cell line (*Homo sapiens*) | K562 | ATCC | | |
| Cell line (African green monkey) | COS-7 | *Gluzman, 1981* | | |
| Sequence-based reagent | gRNAs | This paper | | Check *Supplementary file 1* |
| Sequence-based reagent | Probes, RT-qPCR and PCR primers | This paper | | Check *Supplementary file 2* |
| Software, algorithm | Reditools 2 | GitHub - BioinfoUNIBA/ REDItools2, *Giudice, 2022* | | RNA off-target |
| Software, algorithm | Synthego ICE | Synthego | | InDel for Sanger sequenced data |
| Software, algorithm | EditR | EditR: Edit Deconvolution by Inference of Traces in R ( shinyapps.io) | | Base editing efficiency for Sanger sequenced data |
| Software, algorithm | IGV | Home | Integrative Genomics Viewer ( broadinstitute.org) | | Visualize Aligned data |
| Software, algorithm | CRISPResso-2 | CRISPResso2 (partners.org) | | Base editing efficiency for NGS data |
| Software, algorithm | Snapgene | SnapGene | Software for everyday molecular biology | | gRNA designing |
| Software, algorithm | Benchling | CRISPR gRNA Design Tool | Benchling | | gRNA designing |
| Software, algorithm | FlowJo 10.7.1 | Home | FlowJo, LLC | | FACS data analysis |

*Appendix 1 Continued on next page*

*Appendix 1 Continued*

| Reagent type (species) or resource | Designation | Source or reference | Identifiers | Additional information |
|---|---|---|---|---|
| Software, algorithm | Cas off-finder | CRISPR RGEN Tools (rgenome.net) | | DNA off-target prediction |
| Software, algorithm | Cosmid | CRISPR Target Search (gatech.edu) | | Primer designing for predicted DNA off-targets |
| Software, algorithm | Bowtie 2 | Bowtie 2: fast and sensitive read alignment (sourceforge.net) | | Sequence alignment |
| Software, algorithm | TrimGalore | Babraham Bioinformatics - Trim Galore! | | FastQ files processing |
| Software, algorithm | NFCore RNA Seq pipeline | rnaseq » nf-core | | RNA sequencing pipeline |
| Software, algorithm | Interactive Gene Expression Analysis Kit (iGEAK) | iGEAK! (google.com) | | |
| Software, algorithm | GraphPad Prism 8.0.1 | GraphPad Prism (https://graphpad.com) | RRID:SCR_015807 | |

