## [Editor Report]

This paper describes the innovative use of base editing to mutagenize an enhancer region in the iconic globin locus, demonstrating a new method while also finding a potential novel locus for downstream therapeutic approaches.

---

## [Decision Letter]

**Decision letter after peer review:**

Thank you for sending your article entitled "Identification of novel HPFH-like mutations by CRISPR base editing that elevate the expression of fetal hemoglobin" for peer review at *eLife*. Your article is being evaluated by 2 peer reviewers, and the evaluation is being overseen by a Reviewing Editor and Patricia Wittkopp as the Senior Editor.

The main concerns are the extent and varied nature of the different technical aspects of the paper. Whether these can be addressed in a focused fashion within a reasonable time-period is the question at hand.

*Reviewer #1:*

Disrupting transcriptional regulation of HBG to induce fetal hemoglobin (HbF) expression is a promising therapeutic strategy for sickle cell disease (SCD). To identify novel HBG regulatory elements, Ravi et al., utilized cytosine (CBE) and adenine (ABE) base editors to mutagenize the proximal *HBG* promoter region in HUDEP-2 immortalized human erythroid progenitor cells. The authors were able to achieve successful editing across a number of target sites with several inducing functional upregulation of HbF. Notably, for one ABE target the induction of HBG could be explained by the creation of a consensus binding site for KLF1, although the degree of induction was less than that achieved by disruption of a previously identified BCL11A binding site. These data highlight advantages of using base editors as a mutagenesis tool and extend our current understanding of *HBG* gene regulation which may have future therapeutic relevance in SCD.

Strengths:

Genetic mapping of key regulatory elements within the *HGB* promoter region has been conducted using nuclease-based mutagenesis but is limited by the imprecise repair outcomes of NHEJ and by frequent deletion of the region intervening the highly homologous sequences of the duplicated *HBG1* and *HBG2* loci. The use of base editors is an elegant strategy to overcome these challenges.

Although nuclease-based indel induction can be readily used to disrupt regulatory elements, it is less useful for the introduction of novel transcription factor binding sites unless paired with inefficient homology directed repair approaches. The current study demonstrates a key advantage of base editing which is the ability to efficiently create transcription factor binding sites. This broadens the scope of functional alterations that can be introduced within gene regulatory elements.

In addition to demonstrating that base editors are a useful tool for regulatory element mapping, the current study sheds new light on the regulation of *HBG* expression by resolving five additional regulatory regions located within the *HBG* proximal promoter region that could have therapeutic relevance in SCD.

Weaknesses:

Although the paper successfully identifies five novel regulatory regions within the *HBG* promoter that when modified increase HbF expression, none of these edits induce HbF as effectively as disruption of the previously identified BCL11A binding site. Thus, from a translational standpoint the results do not appear to be a significant advance. However, the substantial variation in editing at each site may be limiting the magnitude of HbF induction.

Variable editing across target sites has been described for both CBE and ABE, so it's not unexpected that it was observed in the current study. However, without data showing the transduction efficiency of the sgRNA vectors (by GFP flow) it is difficult to conclude that the variation is solely a function of the target or target sequence context. At a few of the sites with low genetic editing there is a significant increase in HbF expression, suggesting that if the editing at these sites were improved the functional induction of HbF could be substantial.

The use of lentivirus delivered gRNAs would appear to be amenable to pooled screening, but in the current study the screen was conducted in an arrayed fashion. The ability to conduct a pooled screen could substantially expand the utility of this approach and it would have been very interesting to see how the results from a pooled and arrayed screen compared to one another.

Lastly, while the functional readouts in the current study are compelling and largely correlate with the editing, confirmation in a second system would further inspire confidence in the results. In particular, members of this team and others have carried out editing in primary human CD34+ hematopoietic stem/progenitor cells (HSPC) followed by erythroid differentiation. Confirmation of the identified targets in primary cells with therapeutic relevance would inspire further confidence in the results.

– GFP flow data for the transduction efficiency of sgRNA vectors should be included. This is critical to confirm that the editing variability is due to target site and not transduction efficiency. It would also be helpful to confirm that none of the gRNA sequences contain premature termination sequences as well.

– The nature of the replicates are unclear. In figures 2 and 3 there are no error bars and in the figures with error bars it in not clear the definition of the biological replicates. Were these independent transductions of the cells, or repeat measures of the same transduced populations?

– In the functional validation experiments, were these independent transductions or use of the initial transduced populations? What were the sgRNA transduction rates in the populations used?

– Several previous editing strategies have been used to increase HbF. It would be informative if prior editing strategies were included in this study for comparison.

– Quantification of the EMSA in figure 6c should be included.

– Additional analysis to quantify and more clearly represent the correlation between genome editing and HBG (HbF) induction would be helpful for readers.

– Follow up validation of promising targets using transient transfection approaches would be informative as well.

– Despite low indels the frequency of deletion between *HBG1* and *2* by PCR should be measured and the gel included next to a nuclease control.

– ABE gRNA 03 gave low genetic editing rates but apparently higher HbF induction. it would be very interesting to see how this target behaved if targeted with recently published hyperactive (ABE8e etc) variants. If high editing were achieved it would seem this could be an attractive target for therapy.

– Additional analysis to quantify and more clearly represent the correlation between genome editing and HBG (HbF) induction would be helpful for readers.

– For promising targets, confirmation in primary CD34+ HSPC would increase the impact of the current study.

– The observation of editing outside the spacer window described as novel in line 188 has been previously observed: Webber et al., Nature Comm (https://www.nature.com/articles/s41467-019-13007-6), Arbab et al., Cell (https://www.sciencedirect.com/science/article/abs/pii/S0092867420306322). These studies should be cited and text changed to describe these results as confirmatory.

– Line 221 describing reduction of double positive cells in CBE treated cells. Since results are not significant this statement should be removed.

*Reviewer #2:*

In this study, the Authors use base editors to generate mutations in the promoters of the genes encoding the γ-globin chains of the fetal hemoglobin in an adult erythroid cell line. Several of these mutations are associated with increased γ-globin gene expression, paving the way for the development of therapeutic strategies for β-hemoglobinopathy patients who have been shown to benefit from persistent expression fetal hemoglobin in adulthood. The major strengths of this work are: (i) the discovery of novel mutations associated with elevated feta hemoglobin levels; (ii) the impact of these discovery in the field of gene therapy for β-hemoglobinopathies.

The weakness of this study is the technical quality that can be improved to support the Authors' conclusions.

Remarks to the Authors:

– While many findings of this study are interesting, in the Results section it is not clear which are the specific questions. The results are a list of data that are partially interpreted only in the Discussion. I would suggest the Authors to better explain their findings in the Results section (e.g., comparison of different strategies disrupting repressor binding sites or recruiting transcriptional activators).

– It would be interesting to study if any of the mutations that do not increase HbF, down-regulate γ-globin expression (in cells that do express HbF) to identify binding sites for activators.

– The Authors should better characterize the cell lines expressing base editors. If I understood correctly, these cells are constitutively expressing the base editors (and the gRNAs). Some base editors are known to induce off-targets at DNA and RNA levels. This possibility should be investigated (at least the RNA off-targets) to verify that the major players in HbF regulation are properly expressed by performing a side-by-side comparison of cells expressing the base editors and untransduced cells. In addition, if base editors and gRNA are constitutively expressed, base editing efficiency can change overtime and should be evaluated in the same cells in which the Authors measure HbF expression.

– Evaluation of HbF+ cells (by flow cytometry) and globin mRNAs (by qRT-PCR) should be performed in differentiated cells, as HbF expression changes upon differentiation.

– The Authors should specify if control cells express the base editors and a control gRNA.

– The first screening of gRNAs is performed without replicates (Figure 2 and 3 and related supplementary figures). Furthermore, many of the gRNAs produce very few editing events, thus it is not possible rule out if the target regions are important for γ-globin expression. Therefore, I would exclude from the analysis the gRNAs associated with a low genome editing as these results are confounding and do not give insights in the regulation of γ-globin expression. Maybe Figure 2 and 3 (and related supplementary figures) could be kept as supplementary data.

– Line 178: the -117 mutation (occurring upstream of the gRNA target site) should be reported in Figure 4.

– Interestingly, the Authors observed that the -123/-124 mutations create a binding site for KLF1. However, most of the editing events generate only one of the 2 mutations (figure 4 suppl 2), would individual mutations be sufficient to generate a KLF1 binding site? Can this be tested by performing an EMSA assay with oligos harboring individual mutations? If individual mutations are not generating a KLF1 binding site, would the observed HbF reactivation be justified by the recruitment of KLF1 only to a minor fraction of the alleles harboring the concomitant editing of the 2 nucleotides?

Finally, can the Authors perform ChIP experiments to demonstrate the increase KLF1 recruitment upon editing of the -123/-124 region?

– Lines 311-314: "CBE mediated base conversion (C to T) at positions – 114 and -115 resulted in significantly greater induction of HbF than the multiple A to G nucleotide substitutions at -110, -112, -113, -116 positions made by ABE." I believe that this or similar statements should be supported by the analysis of clonal populations harboring the same number of edited promoters.

– Figure 3-Suppl Figure 1: it would be helpful to plot HbF expression of the negative controls.

– SEM is missing in Figure 4-suppl Figure 1, panels A and C.

– Figure 5-Suppl Figure 1, panel d: why α and β chains are reduced in samples gRNA02 and gRNA 42?

– Figure 5-Suppl Figure 1: Can the Authors comment on the imbalanced Agamma/Ggamma ratio in sample gRNA 42? Could it be due to the potential deletion of the *HBG2* gene? Besides Indels, did the Authors measure the frequency of the 4.9kb deletions in base-edited samples?

– It would be interesting to perform experiments in primary cells to validate these interesting findings.

---

## [Author Response]

Reviewer #1:Disrupting transcriptional regulation of HBG to induce fetal hemoglobin (HbF) expression is a promising therapeutic strategy for sickle cell disease (SCD). To identify novel HBG regulatory elements, Ravi et al., utilized cytosine (CBE) and adenine (ABE) base editors to mutagenize the proximal *HBG* promoter region in HUDEP-2 immortalized human erythroid progenitor cells. The authors were able to achieve successful editing across a number of target sites with several inducing functional upregulation of HbF. Notably, for one ABE target the induction of HBG could be explained by the creation of a consensus binding site for KLF1, although the degree of induction was less than that achieved by disruption of a previously identified BCL11A binding site. These data highlight advantages of using base editors as a mutagenesis tool and extend our current understanding of *HBG* gene regulation which may have future therapeutic relevance in SCD.Strengths:Genetic mapping of key regulatory elements within the *HGB* promoter region has been conducted using nuclease-based mutagenesis but is limited by the imprecise repair outcomes of NHEJ and by frequent deletion of the region intervening the highly homologous sequences of the duplicated *HBG1* and *HBG2* loci. The use of base editors is an elegant strategy to overcome these challenges.Although nuclease-based indel induction can be readily used to disrupt regulatory elements, it is less useful for the introduction of novel transcription factor binding sites unless paired with inefficient homology directed repair approaches. The current study demonstrates a key advantage of base editing which is the ability to efficiently create transcription factor binding sites. This broadens the scope of functional alterations that can be introduced within gene regulatory elements.In addition to demonstrating that base editors are a useful tool for regulatory element mapping, the current study sheds new light on the regulation of *HBG* expression by resolving five additional regulatory regions located within the *HBG* proximal promoter region that could have therapeutic relevance in SCD.

We thank the reviewer for the supportive words and the positive comments about our study.

Weaknesses:Although the paper successfully identifies five novel regulatory regions within the *HBG* promoter that when modified increase HbF expression, none of these edits induce HbF as effectively as disruption of the previously identified BCL11A binding site. Thus, from a translational standpoint the results do not appear to be a significant advance. However, the substantial variation in editing at each site may be limiting the magnitude of HbF induction.

The reviewer brings up an excellent consideration. As the reviewer has rightly pointed out, one of the shortcomings in this study is to obtain equivalent editing efficiency in all the target regions and correlate the extent of editing with HbF levels. To overcome this problem, we have used ABE 8e (a variant with high processivity) to increase the editing efficiency of low editing gRNA with high HbF (Figure 3-Sup figure 4). Finally, we have evaluated a novel target site (gRNA 11) identified from this study with ABE 8e in healthy donor CD34+ HSPCs. As expected, we were able to obtain higher editing efficiency at the target site. Importantly, the corresponding increase in HbF levels was better than the current clinical trial target that disrupts the BCL11A binding site (Figure 4).

Variable editing across target sites has been described for both CBE and ABE, so it's not unexpected that it was observed in the current study. However, without data showing the transduction efficiency of the sgRNA vectors (by GFP flow) it is difficult to conclude that the variation is solely a function of the target or target sequence context. At a few of the sites with low genetic editing there is a significant increase in HbF expression, suggesting that if the editing at these sites were improved the functional induction of HbF could be substantial.

We appreciate the reviewer’s comment. We have now included the analysis of transduction efficiency for all the gRNAs as suggested by the reviewer. The data indicate that the variation in the HbF levels depends on the target site rather than the transduction efficiency (Figure 2 and Figure 3-Sup figure 1). Also, we have evaluated the low editing gRNA with high HbF using the ABE 8e variant, which significantly increased the editing efficiency and the HbF levels (Figure 3-Sup figure 4).

The use of lentivirus delivered gRNAs would appear to be amenable to pooled screening, but in the current study the screen was conducted in an arrayed fashion. The ability to conduct a pooled screen could substantially expand the utility of this approach and it would have been very interesting to see how the results from a pooled and arrayed screen compared to one another.

This is an interesting suggestion. Previous work published by Kim et al., (Genome Res. 2018 Jun; 28(6): 859-868) showed that arrayed CRISPR based screen can be more sensitive than a pooled screen. Their group has pointed out that subtle phenotypic effects may not be observable in a pooled screen when compared to an array screen, as the samples in arrayed screens exhibit clear phenotypic variation even with minor editing. Another study (Raphaella W L So et al., Mol Neurodegener. 2019 Nov 14;14(1):41.) has reported that cells in a pooled culture can negatively affect other cells particularly when they undergo inflammatory response or senescence after editing, this is not the case in arrayed screen as the individual targets are separated.

Lastly, while the functional readouts in the current study are compelling and largely correlate with the editing, confirmation in a second system would further inspire confidence in the results. In particular, members of this team and others have carried out editing in primary human CD34+ hematopoietic stem/progenitor cells (HSPC) followed by erythroid differentiation. Confirmation of the identified targets in primary cells with therapeutic relevance would inspire further confidence in the results.

We appreciate the reviewer’s comment. We have now performed base editing in CD34+ HSPCs and were able to achieve HbF levels at a therapeutically significant level (Figure 4).

– GFP flow data for the transduction efficiency of sgRNA vectors should be included. This is critical to confirm that the editing variability is due to target site and not transduction efficiency. It would also be helpful to confirm that none of the gRNA sequences contain premature termination sequences as well.

We agree entirely with the reviewer and have included the plot for the transduction efficiency of individual gRNAs with their respective editing efficiency at the target site (Figure 2c and d and Figure 3-Sup figure 1f). We also thank the reviewer for bringing the premature termination sequences to our attention. We have verified that there are no premature termination sequences in the gRNAs used in our study.

– The nature of the replicates are unclear. In figures 2 and 3 there are no error bars and in the figures with error bars it in not clear the definition of the biological replicates. Were these independent transductions of the cells, or repeat measures of the same transduced populations?

We apologize for the confusion. We performed the first round of screening for the 41 gRNAs in two different base editor expressing HUDEP-2 cell lines (ABE and CBE) without replicates (data in Figure 2c and d). After performing the screen, the top 8 gRNAs based on HBF expression were taken for further validation in triplicates with independent transduction of cells (Figure 3).

– In the functional validation experiments, were these independent transductions or use of the initial transduced populations? What were the sgRNA transduction rates in the populations used?

Thank you for this comment. The functional validation experiments were performed as three independent transductions of cells. The transduction efficiency of all the gRNAs were presented in Figure 3-Sup figure 1f**.**

– Several previous editing strategies have been used to increase HbF. It would be informative if prior editing strategies were included in this study for comparison.

This is an excellent suggestion. We have compared the editing of previously identified BCL11A binding site with the Cas9 and two different base editors (CBE and ABE) in parallel to evaluate the fetal globin expression in HUDEP-2 cells. Indeed, we observed the higher frequency of large deletions (encompassing *HBG2* gene) with the lower level of G γ chain expression in Cas9 edited cells in comparison with ABE and CBE. These results further validate the conclusion that ABE and CBE are highly efficient in editing the highly homologous regions like γ globin promoter without causing any major deletions. We have added the new figure (Figure 1) and relevant descriptions based upon these results in the revised manuscript.

– Quantification of the EMSA in figure 6c should be included.

Thank you for this comment. As suggested by the reviewer, we have quantified the EMSA bands based on its intensity and included these data in Figure 5d and Figure 5-Sup figure 1b.

– Additional analysis to quantify and more clearly represent the correlation between genome editing and HBG (HbF) induction would be helpful for readers.

We completely agree with the reviewer. We have now represented the correlation between genome editing and HbF induction more clearly using principal component analysis (PCA) in Figure 3i. Among the validated top eight gRNAs in ABE and CBE, gRNA- 2, 10, 11 with ABE and gRNA-2 with CBE resulted in a high target editing efficiency with a corresponding increase in HbF expression. In case of gRNA 42 with CBE, only a modest level of HbF elevation was achieved even with higher editing efficiency. On the other hand, gRNAs -3 and 4 with ABE and gRNAs -10 and 11 with CBE showed higher elevation of HbF levels despite lower base conversion efficiency. Overall, the potential gRNAs include gRNA-2, 3, 4, 10, and 11 with ABE and gRNA-2 with CBE that exhibit high induction of HbF expression.

– Follow up validation of promising targets using transient transfection approaches would be informative as well.

We thank the reviewer for this comment. To this end, we have performed base editing of promising target regions in human CD34+ HSPCs by electroporation of ABE8e mRNA with gRNA-2 or gRNA-11 that targets the BCL11A binding motif at position −118 to −114 position and the putative KLF1 consensus motif at -123 and -124 position in the *HBG1*/*HBG2* gene promoters respectively. We cultured the base- edited CD34+ HSPCs under erythroid differentiation conditions and determined the effect of base editing on HbF expression, erythroid differentiation, and enucleation. We have now included the comprehensive analysis of the base editing healthy donor CD34+ HSPCs by transient transfection in the revised manuscript (Figure 4).

– Despite low indels the frequency of deletion between *HBG1* and *2* by PCR should be measured and the gel included next to a nuclease control.

Thank you for the thoughtful comment. As per the reviewer’s suggestion, we have performed the 4.9kb deletion analysis in the edited samples by using qRT PCR as previously reported by André Lieber’s group (Chang Li et al., Blood. 2018 Jun 28;131(26): 2915-2928.). We have validated the 4.9kb large deletion frequency (encompassing *HBG2* gene) for ABE, CBE and Cas9 in HUDEP-2 cell lines targeting the BCL11A binding site at the highly homologous *HBG* promoter. The Cas9 edited cells resulted in higher frequency of 4.9 kb deletion as anticipated. Interestingly, we have observed the deletions in the base edited cells but significantly lower when compared to Cas9 (Figure 1c). To extend this finding, we measured the frequency of the larger deletion for the other potential target sites that resulted in lower level of *HBG2* chain expression (Figure 3-Sup figure 3e and k) and also in base edited CD34+ HSPCs cells (Figure 4c).

– ABE gRNA 03 gave low genetic editing rates but apparently higher HbF induction. it would be very interesting to see how this target behaved if targeted with recently published hyperactive (ABE8e etc) variants. If high editing were achieved it would seem this could be an attractive target for therapy.

Thank you for this comment. We have now evaluated the gRNA-3 which resulted in higher induction of HbF with low editing efficiency using the hyperactive variant ABE8e and subsequently analyzed for the further possible increase in HbF expression by increasing the editing efficiency in HUDEP-2 cells. The ABE8e variant has greatly increased the base editing efficiency at the target site and resulted in higher levels of HbF induction with reduced frequency of the larger deletion in comparison with ABE7.10 (Figure 3-Sup figure 4d). This result suggests that adenine base editing of the *HBG1* and *HBG2* promoters to create the -175 T>C change with ABE 8e is a potential strategy for the therapeutic induction of fetal globin level and treatment for sickle disease and β thalassemia.

– Additional analysis to quantify and more clearly represent the correlation between genome editing and HBG (HbF) induction would be helpful for readers.– For promising targets, confirmation in primary CD34+ HSPC would increase the impact of the current study.

We thank the reviewer for the great suggestion. As detailed above in Response 1.7, we analyzed the therapeutic potential of novel targets (gRNA-11) identified from this study on induction of γ globin expression by base editing of CD34+ HSPCs from a healthy donor (Figure 4). Base editing of the -123 region of *HBG* promoter resulted in a therapeutic level of induction of HbF in erythroblasts derived from human CD34+ HSPCs, without having any detrimental effects on erythroid differentiation and enucleation. Notably, the level of HbF induction was more pronounced with the installation of novel -123 cluster HPFH like mutations than the creation of -115 cluster HPFH mutation which disrupts the BCL11A binding site. The results indicate that base editing at the -123 region of *HBG* promoter may offer a new therapeutic approach for the treatment of β-hemoglobinopathies.

– The observation of editing outside the spacer window described as novel in line 188 has been previously observed: Webber et al., Nature Comm (https://www.nature.com/articles/s41467-019-13007-6), Arbab et al., Cell (https://www.sciencedirect.com/science/article/abs/pii/S0092867420306322). These studies should be cited and text changed to describe these results as confirmatory.

We apologize for the mistake. We have now added the relevant citations and rephrased the sentence in the Results section.

– Line 221 describing reduction of double positive cells in CBE treated cells. Since results are not significant this statement should be removed.

We agree entirely with the reviewer and have removed the statement in the revised manuscript*.*

Reviewer #2:In this study, the Authors use base editors to generate mutations in the promoters of the genes encoding the γ-globin chains of the fetal hemoglobin in an adult erythroid cell line. Several of these mutations are associated with increased γ-globin gene expression, paving the way for the development of therapeutic strategies for β-hemoglobinopathy patients who have been shown to benefit from persistent expression fetal hemoglobin in adulthood. The major strengths of this work are: (i) the discovery of novel mutations associated with elevated feta hemoglobin levels; (ii) the impact of these discovery in the field of gene therapy for β-hemoglobinopathies.

We thank the reviewers for highlighting the importance of our study, as we also believe that it will advance the field of gene therapy for β-hemoglobinopathies.

The weakness of this study is the technical quality that can be improved to support the Authors' conclusions.

We appreciate the reviewer’s constructive comment. As per the reviewer’s suggestion, we have worked on the technical quality of our study and manuscript to further improve and support our conclusions.

Remarks to the Authors:– While many findings of this study are interesting, in the Results section it is not clear which are the specific questions. The results are a list of data that are partially interpreted only in the Discussion. I would suggest the Authors to better explain their findings in the Results section (e.g., comparison of different strategies disrupting repressor binding sites or recruiting transcriptional activators).

We thank the reviewer for the great suggestion. We agree entirely with the reviewer and acknowledge that in the first version of the manuscript we did not adequately interpret the results. Prompted by the comment of the reviewers we have reworked the current version of the manuscript in the Results section to include a detailed explanation of the findings and interpreted the results more clearly as suggested.

– It would be interesting to study if any of the mutations that do not increase HbF, down-regulate γ-globin expression (in cells that do express HbF) to identify binding sites for activators.

It is a very interesting perspective that the reviewer has pointed out. To examine this observation, we have selected the gRNAs that have higher editing efficiency at the target site with the decreased expression of γ-globin from our preliminary screen. These gRNAs were screened in K562 cell line (which has higher basal level of HbF) with the base editors to identify the potential activator binding sites in the *HBG* promoter. The percentage of HbF positive cells remains unaltered after editing suggesting that the targeted regions did not have binding sites for transcriptional activators. This new data is now discussed in revised version the manuscript (Figure 2-Sup figure 3).

– The Authors should better characterize the cell lines expressing base editors. If I understood correctly, these cells are constitutively expressing the base editors (and the gRNAs). Some base editors are known to induce off-targets at DNA and RNA levels. This possibility should be investigated (at least the RNA off-targets) to verify that the major players in HbF regulation are properly expressed by performing a side-by-side comparison of cells expressing the base editors and untransduced cells. In addition, if base editors and gRNA are constitutively expressed, base editing efficiency can change overtime and should be evaluated in the same cells in which the Authors measure HbF expression.

We thank the reviewer for the excellent suggestion. We used the constitutively expressing base editor along with the gRNA for our experiments. We have now conducted the transcriptome profiling of base editor stable cell line (both ABE7.10 and BE3) and compared with the wild type HUDEP-2 cells. The data presented in the new figure (Figure 1-Sup figure 1 (b)) shows that the gene expression profiles are not altered and exhibit a significant correlation with the wild type HUDEP-2 cells. We have also characterized the base editor stable cell lines for the editing efficiency at different time points (in days) and its effect on HbF expression using the known gRNA-2 (targeting BCL11A binding site). The editing efficiency and HbF levels in both ABE and CBE increases over time with no discernible effect on erythroid differentiation (Figure 1-Sup figure 1 (c-h)). As the reviewers have suggested, we have also investigated the possible DNA and RNA off target effects and analyzed the expression of major molecular targets that are involved in HbF regulation in the base edited samples. Transcriptome wide RNA sequencing on ABE and CBE stables with gRNAs 2 or 11 confirms that the distribution frequency of A-to- I (in ABE) or C-to-U (in CBE) conversion across the base edited samples were very similar to that of the parental stable cell line (Figure 6b-e). The Cas-dependent DNA off target were also validated, we were not able to observe any significant undesired off target effects despite higher on target editing efficiency (Figure 6a). We performed the differential expression analysis for the 34 selected genes that are involved in globin regulation and observed no significant difference in base edited cells compared to the control (Figure 6-Sup figure 1).

– Evaluation of HbF+ cells (by flow cytometry) and globin mRNAs (by qRT-PCR) should be performed in differentiated cells, as Hb expression changes upon differentiation.

Thank you for this comment. We agree with the reviewer that HbF expression will change upon erythroid differentiation. As the reviewer suggested, we have determined the level of globin mRNA by qRT-PCR and HbF positive cells by flow cytometry for the top 8 targets in the ABE and CBE edited cells after erythroid differentiation (Figure 3c and d, Figure 3-Sup figure 3b and h ). The number of HbF positive cells in differentiated erythroid cells was slightly higher than that of the undifferentiated cells. The relative expression of γ globin in the erythroid differentiated samples followed a similar trend to that of the undifferentiated sample.

– The Authors should specify if control cells express the base editors and a control gRNA.

Thank you for this comment. The control cells used in our study constitutively express the base editors and the control gRNA (with scrambled sequence) in HUDEP2 cells. In case of CD34+ HSPCs, we used the gRNA targeting the AAVS1 site (a safe harbour locus) along with base editor mRNA as a control. We have added the relevant information in the revised manuscript as suggested by the reviewer.

– The first screening of gRNAs is performed without replicates (Figure 2 and 3 and related supplementary figures). Furthermore, many of the gRNAs produce very few editing events, thus it is not possible rule out if the target regions are important for γ-globin expression. Therefore, I would exclude from the analysis the gRNAs associated with a low genome editing as these results are confounding and do not give insights in the regulation of γ-globin expression. Maybe Figure 2 and 3 (and related supplementary figures) could be kept as supplementary data.

We thank the reviewer for the suggestion. We agree entirely with the reviewer that the target region of many of the low editing gRNAs might have a possible role in γ globin regulation. Therefore, we excluded the gRNAs-37,38,7,18,19 and 29 from ABE and gRNAs-1,35,37,6,7,13 and 19 from CBE from the analysis because of the lower editing efficiency (<10%) at the target site. We have moved the figure 2 to the supplementary figures (Figure 2-Sup figure 1a and c) as suggested by the reviewer. Figure 3 was kept in main figure (as Figure 2) as it is to represent the level of HbF induction with the total editing efficiency for all the gRNAs from the primary screening.

– Line 178: the -117 mutation (occurring upstream of the gRNA target site) should be reported in Figure 4.

We appreciate the reviewer’s suggestion on including the base conversion events that occurred upstream of the protospacer target site. We have now represented all these mutations in the new figure (Figure 3-Sup figure1b) as suggested.

– Interestingly, the Authors observed that the -123/-124 mutations create a binding site for KLF1. However, most of the editing events generate only one of the 2 mutations (figure 4 suppl 2), would individual mutations be sufficient to generate a KLF1 binding site? Can this be tested by performing an EMSA assay with oligos harboring individual mutations? If individual mutations are not generating a KLF1 binding site, would the observed HbF reactivation be justified by the recruitment of KLF1 only to a minor fraction of the alleles harboring the concomitant editing of the 2 nucleotides?Finally, can the Authors perform ChIP experiments to demonstrate the increase KLF1 recruitment upon editing of the -123/-124 region?

We thank the reviewer for the excellent suggestion. We have performed a new EMSA to determine the effect on KLF1 binding to a probe containing the individual and combination of -123/ -124 mutations along with the wild type probe. The results indicate that the combination of -123 T>C and -124 T>C mutation (but not with the -123 T>C or -124 T>C individual mutation) is required for the KLF1 binding to the *HBG* promoter in vitro (Figure 5c-d, Figure 5-Sup figure 1a and b). To substantiate this finding, we improved the editing efficiency at the target site with the hyperactive variant of ABE (ABE 8e) and gRNA-11 which results in the higher proportion of -123 T>C and -124 T>C combination mutations with substantial increase in HbF expression in the CD34+ HSPCs (>90% base substitution at 123 T>C and -124 T>C position with >90% HbF+ cells) (Figure 4b and e). The ChIP results were not conclusive. They did not show a significant enrichment of KLF-1 at the 123 T>C and -124 T>C mutated region of *HBG* promoter when compared to an arbitrarily chosen negative control (VEGFA) but did exhibit slightly better enrichment compared to the WT clones (Figure 5-Sup figure 1c). It should be noted that as seen in the EMSA the KLF1 binding is at best weak and may be below the level of detection by ChIP in these experiments. Future investigations would be required to confirm that KLF1 binding to this site is the main in vivo mechanism of -123 T>C and -124 T>C HPFH driven up-regulation of γ globin. We have worded our manuscript carefully to make this point clear.

– Lines 311-314: "CBE mediated base conversion (C to T) at positions – 114 and -115 resulted in significantly greater induction of HbF than the multiple A to G nucleotide substitutions at -110, -112, -113, -116 positions made by ABE." I believe that this or similar statements should be supported by the analysis of clonal populations harboring the same number of edited promoters.

We respectfully suggest that the effect of individual mutations at the -115 clusters of *HBG* promoter on γ globin regulation are extensively characterized by previous publications (Martyn et al., Nature genetics 2018., Liu N et al., Cell 2018). Further, we believe that the change in HbF induction by ABE and CBE are mainly due to the variation in the binding affinity of BCL11A towards the mutated core TGACCA motif. The previous publication by Yang et al., (Cell Research (2019) 29:960– 963) proves that the nucleotides at -114 and -115 positions are very important for BCL11A binding when compared to other nucleotides at the -115 region of *HBG* promoter, which is highly consistent with our observations of CBE mediated base conversion at – 114 and -115 position.

– Figure 3-Suppl Figure 1: it would be helpful to plot HbF expression of the negative controls.

Thank you for this comment. We have included the HbF expression of the negative control and presented in the new updated Figure 2 c and d (previously Figure 3-Sup figure 1) of the revised version of manuscript as suggested by the reviewer.

– SEM is missing in Figure 4-suppl Figure 1, panels A and C.

Thank you for this comment. We have included the SEM for the updated new Figure 3a and b (previously Figure 4-Sup figure 1) as suggest by the reviewers.

– Figure 5-Suppl Figure 1, panel d: why α and β chains are reduced in samples gRNA02 and gRNA 42?

This is a great question. We believe based on the previous report that the higher induction of γ globin chain is shown to downregulate β globin chain expression thereby maintaining the α to β-like globin chain ratio (Huang P et al., Genes Dev. 2017). They have indicated that in fetal stages/HPFH condition, the LCR directly interacts with γ globin promoter for its regulation. Therefore, the LCR will not be able to access β globin promoter which results in reduced expression of β globin chain. In case of α globin, we are not entirely sure about the mechanism, but previous reports have shown that induction of HbF using inducers have down regulated α globin chain expression (Khamphikham P et al., Br J Haematol 2020).

– Figure 5-Suppl Figure 1: Can the Authors comment on the imbalanced Agamma/Ggamma ratio in sample gRNA 42? Could it be due to the potential deletion of the HBG2 gene? Besides Indels, did the Authors measure the frequency of the 4.9kb deletions in base-edited samples?

We thank the reviewer for bringing up an excellent point. We are not entirely sure about the reason for variation in A γ and G γ expression in gRNA 42. As reviewers suggested, we have now performed the qRT-PCR analysis on CBE stables cells transduced with gRNA-42 to ensure that the difference in γ chain expression is due to 4.9kb large deletion. We did not observe the large deletion in gRNA 42 edited cells (Figure 3-Sup figure 3k). Thus, the decrease in G γ chain expression is independent of the large deletion and might be due to the biased expression of the γ globin.

– It would be interesting to perform experiments in primary cells to validate these interesting findings.

We thank the reviewers for the great suggestion. As detailed above in Response 1.7, we performed the base editing to introduce novel -123 cluster mutations at the *HBG* promoter in heathy donor CD34+HSPCs. Targeted adenosine base editing at the -123 region of the *HBG* promoter induced robust γ globin expression (than the base editing at the BCL11 binding site) in CD34+ HSPC-derived human erythroblasts, without having any detrimental effect on erythroid differentiation and maturation (Figure 4).